# Structure of a Holliday junction complex reveals mechanisms governing a highly regulated DNA transaction

**Gurunathan Laxmikanthan[1,2], Chen Xu[3], Axel F Brilot[3], David Warren[1,2], Lindsay Steele[1,2], Nicole Seah[1,2], Wenjun Tong[1,2], Nikolaus Grigorieff[3,4]\*, Arthur Landy[1,2]\*, Gregory D Van Duyne[5]\***

[1]Department of Molecular Biology, Cell Biology, and Biochemistry, Brown University, Providence, United States; [2]Division of Biology and Medicine, Brown University, Providence, United States; [3]Department of Biochemistry, Rosenstiel Basic Medical Sciences Research Center, Brandeis University, Waltham, United States; [4]Janelia Research Campus, Howard Hughes Medical Institute, Ashburn, United States; [5]Department of Biochemistry and Biophysics, Perelman School of Medicine, University of Pennsylvania, Philadelphia, United States

**Abstract** The molecular machinery responsible for DNA expression, recombination, and compaction has been difficult to visualize as functionally complete entities due to their combinatorial and structural complexity. We report here the structure of the intact functional assembly responsible for regulating and executing a site-specific DNA recombination reaction. The assembly is a 240-bp Holliday junction (HJ) bound specifically by 11 protein subunits. This higher-order complex is a key intermediate in the tightly regulated pathway for the excision of bacteriophage λ viral DNA out of the *E. coli* host chromosome, an extensively studied paradigmatic model system for the regulated rearrangement of DNA. Our results provide a structural basis for pre-existing data describing the excisive and integrative recombination pathways, and they help explain their regulation.

\*For correspondence: niko@
grigorieff.org (NG); arthur_landy@
brown.edu (AL); vanduyne@mail.
med.upenn.edu (GDVD)

**Competing interest:** See
page 20

**Reviewing editor:** Sjors HW
Scheres, Medical Research
Council, United Kingdom

## Introduction

The rearrangement of DNA, either by homologous or site-specific recombination and transposition, is a fundamental feature of evolution, genetic variation, and gene regulation; among such pathways, the integration and excision of bacteriophage λ into and out of the *E. coli* chromosome is one of the most thoroughly characterized. Nevertheless, even for this pathway, it has not been possible until now to visualize the entire functional structure of the recombinogenic complex, despite the wealth of genetic, biochemical, functional and structural information accumulated by many laboratories over the past 50 years.

The site-specific recombinase (Int) encoded by bacteriophage λ is the archetypical member of the tyrosine recombinase family, whose members carry out such diverse functions as chromosome segregation, chromosome copy number control, gene expression, conjugative transposition, gene dissemination, and viral integration and excision (*Craig et al., 2015*; *Jayaram et al., 2015*; *Landy, 2015*). All tyrosine recombinases use the same isoenergetic phosphoryl transfer chemistry and sequential strand exchange mechanism to execute DNA rearrangements via the formation and resolution of a transient four-way DNA junction, or Holliday junction (HJ), recombination intermediate (*Hsu and Landy, 1984*; *Kitts and Nash, 1988a*). Although these junctions are common intermediates in many different pathways responsible for rearranging genetic material in evolution, heredity, and gene

**eLife digest** Some viruses can remain dormant inside an infected cell and only become active when conditions are right to multiply and infect other cells. Bacteriophage λ is a much-studied model virus that adopts this lifecycle by inserting its genetic information into the chromosome of a bacterium called *Escherichia coli*. Certain signals can later trigger the viral DNA to be removed from the bacterial chromosome, often after many generations, so that it can replicate and make new copies of the virus. Specific sites on the viral and bacterial DNA earmark where the virus's genetic information will insert and how it will be removed. Remarkably, each of these two site-specific reactions (i.e. insertion and removal) cannot be reversed once started, and their onset is precisely controlled.

These reactions involve a molecular machine or complex that consists of four enzymes that cut and reconnect the DNA strands and seven DNA-bending proteins that bring distant sites closer together. Despite decades of work by many laboratories, no one had provided a three-dimensional image of this complete molecular machine together with the DNA it acts upon.

Now, Laxmikanthan et al. reveal a three-dimensional structure of this machine with all its components by trapping and purifying the complex at the halfway point in the removal process, when the DNA forms a structure known as a "Holliday junction". The structure was obtained using electron microscopy of complexes frozen in ice. The structure answers many of the long-standing questions about the removal and insertion reactions. For example, it shows how the DNA-bending proteins and enzymes assemble into a large complex to carry out the removal reaction, which is different from the complex that carries out the insertion reaction. It also shows that the removal and insertion reactions are each prevented from acting in the opposite direction because the two complexes have different requirements.

These new findings improve our understanding of how the insertion and removal reactions are precisely regulated. Laxmikanthan et al.'s results also serve as examples for thinking about the complicated regulatory machines that are widespread in biology.

expression, they have been particularly well-studied in the tyrosine recombinase family of reactions (*Gopaul et al., 1998*; *Jayaram et al., 2015*; *Van Duyne, 2015*).

In contrast to two other well-studied and highly exploited family members, Cre and FLP, where the entire bidirectional recombination reaction is executed by a recombinase tetramer (*Jayaram et al., 2015*; *Van Duyne, 2015*), Int carries out two opposing reactions that are both highly directional and tightly regulated. Each reaction requires the assembly of a unique 400 kDa multi-protein complex using accessory DNA-bending proteins and an overlapping ensemble of accessory binding sites. In response to a variety of physiological and environmental signals one complex is assembled to carry out unidirectional integrative recombination between "*att*" sites on the viral and bacterial chromosomes, while the other complex is formed to carry out unidirectional excisive recombination between *att* sites flanking the integrated viral chromosome ([*Seah et al., 2014*; *Tong et al., 2014*]; see *Figures 1* and *2*).

Int is a heterobivalent DNA binding protein that binds to high affinity 'arm-type' DNA sites via its small amino-terminal domain (NTD) and more weakly to 'core-type' DNA sites, where DNA cleavage and ligation take place. Int binds to core sites via a central core binding domain (CB) and a C-terminal catalytic domain (CAT); the latter two domains are referred to here as the CTD (see *Figure 2*; *Moitoso et al., 1989*). The Int subunits within a recombinogenic complex bind and bridge arm- and core-type DNA sites in patterns determined by the accessory DNA bending proteins IHF, Xis, and Fis. Differential occupancy of the 16 protein binding sites on the 240 bp *att* site generates two overlapping ensembles that differentiate the integrative and excisive recombination pathways ([*Seah et al., 2014*; *Tong et al., 2014*]; (see *Figure 1*).

Using techniques described previously (*Figure 3*), we trapped and purified the multi-protein HJ complex of excisive recombination and determined its structure at 11 Å resolution using single particle electron cryo-microscopy (cryo-EM). We then used the EM density, known protein-DNA crystal and NMR structures, and the known Int-mediated arm-core bridges to build an atomic model of the

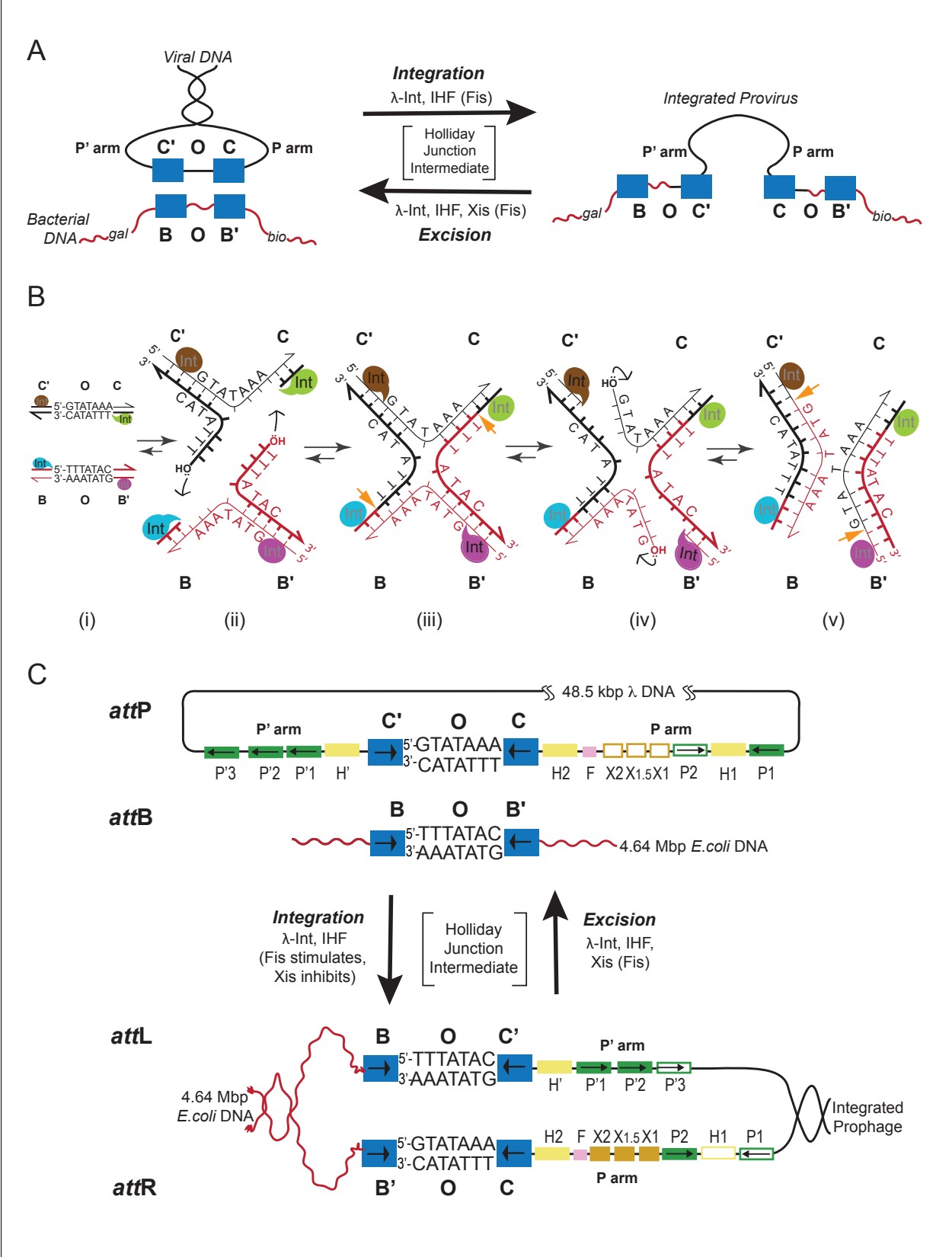

**Figure 1.** Integration and excision of the λ viral chromosome into and out of the bacterial host chromosome is highly directional and tightly regulated. (A) Formation of the integrated prophage. In those infected cells where the decision has been made not to replicate the viral DNA, the circularized

*Figure 1 continued on next page*

*Figure 1 continued*

supercoiled viral DNA (black lines) is inserted into the bacterial chromosome (curly red lines) at a specific site (called *att*B or BOB') located between the gal and bio genes (*Campbell, 1963*). This insertion, integrative recombination, involves cutting the viral DNA at a specific site (called *att*P or P'C'OCP) and joining these cut ends to the cut ends of the bacterial chromosome in *att*B. The cutting, recombining, and resealing of viral and bacterial DNA generates new DNA sequences forming the junctions between bacterial and integrated viral DNA. These junction sequences, called *att*L (BOC'P') and *att*R (PCOB') on the left and right respectively, are themselves substrates for the cutting, recombining, and resealing reactions that will, sometime in the future, remove (excise) the viral DNA from the host chromosome and thereby regenerate the viral *att*P and bacterial *att*B sequences. The integrated provirus chromosome is stably inherited, with almost all of its genes repressed, for many bacterial generations. Upon instigation by the appropriate physiological signals, the viral chromosome is excised, replicated, and inserted into viral particles which are released into the environment. The integrative reaction requires the virally-encoded integrase protein (Int) and the bacterial accessory protein Integration Host Factor (IHF). The excisive reaction additionally requires the virally encoded accessory protein Xis (which also inhibits the integrative reactions). Both reactions are stimulated by the bacterial accessory protein Fis. (See Panel **C** for DNA binding sites of all the proteins and their respective roles in each reaction). Both reactions proceed through a four-way DNA junction called a Holliday junction (HJ) (see Panel **B** for details of these DNA strand exchanges) (reviewed in [*Landy, 2015*]). (B) The Holliday junction intermediate. Cutting, recombining, and resealing DNA during recombination proceeds by two pairs of sequential single-strand DNA exchanges that are staggered by seven base pairs that are identical in all four *att* sites; they are referred to as the 'overlap' region (O). The molecular details of this recombination are common to all reactions catalyzed by the large family of tyrosine recombinases (except for the size of the O regions and the order of strand exchanges), as first characterized for λ Int, Cre, Flp, and XerC/D (reviewed in [*Van Duyne, 2015*]). It proceeds in the absence of exogenous energy via the formation of high-energy covalent 3'phospho-tyrosine intermediates in the active site of each Int protein. Illustrated here is the pathway for integrative recombination; it would be identical for the excisive reaction but the substrates (left panel) would be *att*L and *att*R, leading to *att*P and *att*B products (right panel). Viral and bacterial DNAs are denoted as in panel A. (i) The *att*P (C'OC) and *att*B (BOB') sites are aligned anti-parallel with respect to their identical overlap sequences. (ii) The first pair of exchanges, always at the **C** (green) and **B** (blue) core sites, is initiated by formation of 3' phospho-tyrosine linkages (with Tyr342) and 5' OH termini. (iii) The 5' OH terminus generated by the cleavage at C attacks the phospho-tyrosine linkage at B to regenerate a new B–C' strand (orange arrow head). Concomitantly, the 5' OH from the cleavage at C attacks the phospho-tyrosine linkage at B to regenerate a new B'–C strand (orange arrow head). Together, this pair of single strand exchanges, at one boundary of the overlap region, forms the four-way DNA (HJ) intermediate. (iv) A similar pair of single strand cleavage, exchange, and resealing reactions is executed by the Ints at **C'** (brown) and **B'** (purple) on the other side of the overlap region. (v) As a result of the two sequential pairs of single-strand exchanges, all four DNA strands have new junction sequences and the HJ is resolved to recombinant products *att*L (C'OB) and *att*R (COB'). C) Additional complexity, in the P' and P arms, confers regulation and directionality to the λ Int reaction. In contrast to the 'simple' Cre and Flp tyrosine recombinases, λ Int (and more than a thousand viral cousins in the public data bases) has two DNA binding domains, as shown in *Figure 2*. The carboxy-terminal domain of Int (CTD) binds at the four sites of DNA cleavage (called core-type sites; blue boxes) and catalyzes the chemistry of DNA cleavage and rejoining; this domain and the core-type sites (B, B', C' and C) are analogous, and very similar, to the Cre and Flp enzymes and their respective DNA targets sites (see panel B). In λ Int an additional small DNA binding domain at the amino terminus (NTD) binds with high affinity to a different family of DNA sequences (arm-type sites) (green boxes) distant from the sites of DNA cleavage and located in the P' and P arms of viral DNA (adjacent to C' and C, respectively). To enable the ('hetero-bivalent') Int to bind simultaneously to both of its DNA targets the core- and arm-type DNA sites are interposed by binding sites for the accessory DNA bending proteins, IHF (yellow boxes, H1, H2, and H'), Xis (gold boxes, X1, X1.5, X2), and Fis (magenta). The DNA bending proteins bring the core- and arm-type sites into close proximity and also serve as essential elements in forming the large multi-protein recombination complexes. Two distinct but overlapping ensembles of binding sites are employed (solid boxes) to generate either the integrative or excisive recombinogenic complexes (reviewed in [*Landy, 2015*]). (For the patterns of Int bridging between core- and arm-type sites in each of the complexes see *Figure 2*).

intact functional complex. The experimental structure for the excisive complex then allowed us to construct a model for the integrative complex. The results provide a structural basis for understanding how these complex DNA recombination machines function and how they are so tightly regulated.

## Results and discussion

### Electron cryo-microscopy structure determination

Stable HJ complexes were trapped by in vitro recombination between *att*L and *att*R partners whose respective seven bp overlap sequences were designed to generate a fully paired HJ overlap sequence that would create mismatched base pairs upon reversal or progression of the recombination reaction (*Matovina et al., 2010*; *Tong et al., 2014*). Following brief crosslinking with 0.0035% glutaraldehyde, the complexes were purified by two cycles of sucrose gradient centrifugation and either frozen in liquid nitrogen or immediately prepared for cryo-EM. The complexes were analyzed for purity and homogeneity by native gel electrophoresis (*Figure 3A*) and negative staining electron

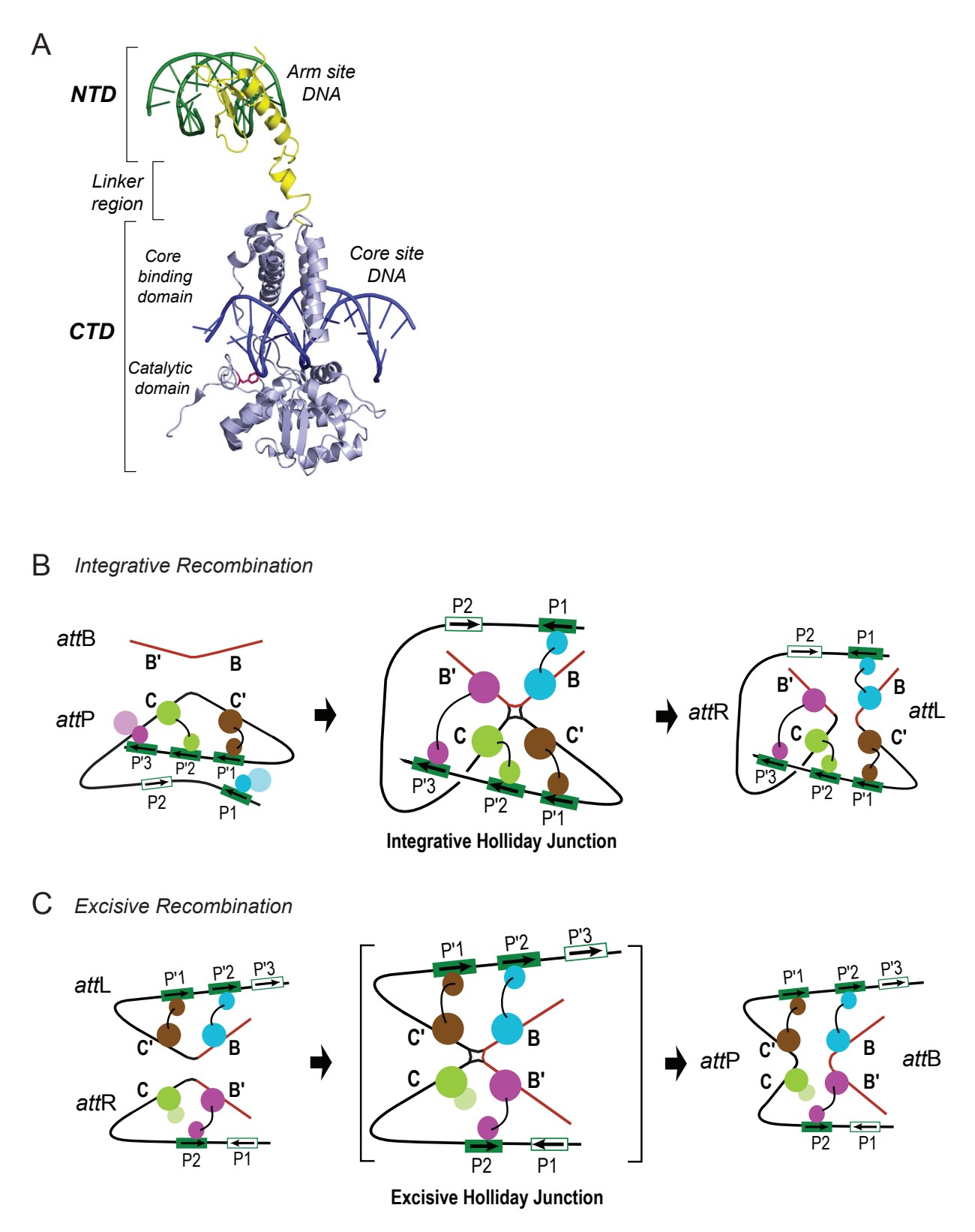

**Figure 2.** Patterns of specific Int bridges between core- and arm-type DNA binding sites. (**A**) The amino-terminal domain (NTD) of Int (yellow) binds to one of the five arm-type DNA sites (green) via a helix-turn-helix recognition motif. It is joined via a flexible 10 residue linker region to the carboxy-
*Figure 2 continued on next page*

*Figure 2 continued*

terminal domain (CTD) (light blue), which is composed of two smaller domains (the core-binding (CB) and catalytic domains) joined by a 12 residue linker. The CTD binds to core-type DNA sites (blue) via a C-clamp motif; the distal domain of the C-clamp contains the nucleophile Tyr-342 (red) and those residues comprising the catalytic site for DNA cleavage and ligation. The CTD is analogous, and very similar in structure and function, to the well-studied monovalent recombinases, Cre, Flp, and XerC/D. (B) Integrative recombination depends upon four Int bridges, as determined previously (*Tong et al., 2014*). In the *att*P and *att*B substrates (left-most panel) only two of the four bridges are formed prior to synapsis; the accessory DNA bending proteins have been omitted for clarity (see *Figure 1C*). The coloring scheme for the Ints bound to different arm-type sites is the same as that used in all of the following figures. (C) Excisive recombination involves a different pattern of Int bridges (*Tong et al., 2014*) and, as shown in *Figure 1C*, a different ensemble of DNA bending proteins. The HJ intermediate of this reaction (brackets) was purified and used for single particle cryo-electron microscopy, as described in the text.

microscopy. For details of the preparation and characterization of the complexes, see (*Tong et al., 2014*).

We selected 66,033 particles from 1359 images (for example, *Figure 3B*) and obtained two-dimensional (2D) class averages using the ISAC procedure (*Yang et al., 2012*) after pixel binning (effective pixel size 5.6 Å). Following visual inspection of the class averages, we selected 52 classes that showed clear features, such as DNA loops and crosses (*Figure 3C*), and calculated initial maps using EMAN2 (*Tang et al., 2007*). The highest-scoring map was used to initialize particle alignment and three-dimensional (3D) classification in FREALIGN (*Lyumkis et al., 2013*). All particles were aligned against the EMAN2 map using data limited to 60 Å resolution. Starting with reconstructions calculated from six randomly selected subsets of the data, particles were 3D-classified into six classes and their alignments refined in 40 iterations in which the resolution limit was increased in regular steps to a final limit of 20 Å. The reconstruction with the highest average particle score was then

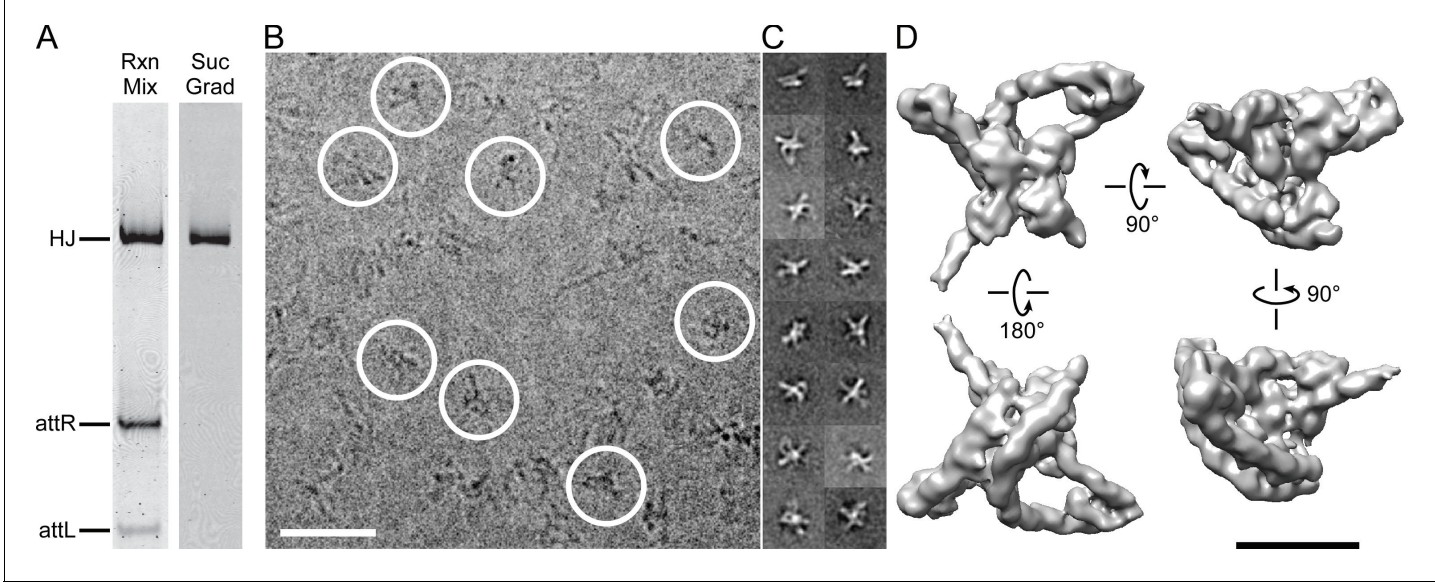

**Figure 3.** Isolation and single particle electron cryo-microscopy of the Holliday junction recombination intermediate. (A) Native polyacrylamide gel electrophorograms of the recombination reactions prior to and after sucrose gradient purification. Recombination reactions between *att*L and *att*R partners designed to trap the HJ intermediate (see *Figure 2*) were treated with 0.0035% glutaraldehyde for 10 min, quenched with 0.3 M glycine and concentrated by ultrafiltration. The concentrated 100 μL samples were loaded onto a 2 mL sucrose gradient (22%–40% sucrose, 10 mM Tris, pH 8, 1 M Betaine) and centrifuged for 16 hr at 4°C. After examining each of the 100 μl fractions by native PAGE the purest fractions were pooled, concentrated, and subjected to a second round of centrifugation. The samples were either immediately frozen in liquid nitrogen for storage or used for plunging grids (see Materials and methods for details). (B) Electron micrograph of ice-embedded Holliday junction complexes. Some complexes used for further processing are encircled. Scale bar = 300 Å. (C) Some 2D class averages obtained from 66,033 particles that were selected from 1359 micrographs. Different views of the complex are apparent. Some of the views clearly show loops, other crosses. (D) Different views of the 3D reconstruction of the Holliday junction complex at 11 Å resolution calculated from 10,956 particles and sharpened with a B-factor of -2500 Å2 (see Materials and methods for details). Scale bar = 100 Å.

used to initialize a second round of alignment and 3D classification, starting again with a resolution limit of 60 Å and increasing the limit in steps to a final 20 Å. Due to the substantial heterogeneity of the sample – only 17% of the particles ended up in the final reconstruction (*Figure 3D*) – we performed six such refinement/classification rounds, always initializing the next round with the reconstruction of the previous round that had the highest average particle score. The highest scoring reconstructions from each round, as well as the initial and final maps are shown in *Figure 4A*. Lower-scoring reconstructions showed some of the features of the highest-scoring reconstruction but were missing density in the DNA loop and/or the Int tetramer regions. This suggests that the heterogeneity is the result of particle distortions and instabilities that often arise during the grid preparation procedure, for example when particles interact with the air/water interface (*Cheng et al., 2015*). For the final map, we performed an additional 50 iterations of refinement and 3D classification using a resolution limit of 18 Å. The last three iterations used data with an effective pixel size of 2.8 Å. The best reconstruction contained 10,956 particles and reached a resolution of about 11 Å (*Figure 4B and C*).

## Construction of an excisive HJ complex EM model

We began by placing a complex containing the core-binding and catalytic domains of an integrase tetramer bound to core site HJ DNA into the EM envelope. This core complex was derived from a crystal structure of full-length integrase bound to HJ DNA and short oligonucleotides containing arm site sequences (*Biswas et al., 2005*). The assembly fit well as a rigid body, indicating that the EM and crystal structures are similar in this region. Structural models for the protein-bound P and P' arms were assembled as previously described (*Seah et al., 2014*), except that the Fis dimer (which is not present in the purified HJ complex used here) was omitted from the P arm model. Details of the arm constructions are provided in Materials and methods. The pre-assembled P and P' arms were spliced onto the core Int-HJ complex while maintaining standard B-DNA twist and fit into the EM envelope by applying small DNA roll angles ( ± 1°) in the regions outside of the IHF, Xis, and Int-NTD footprints. Flexibility in the IHF-induced DNA bends at H' and H2 was also allowed by testing small in-plane and out of plane deviations from the IHF/H'-DNA crystal structure (*Rice et al., 1996*).

Two regions in the resulting structure could not be confidently modeled based on existing crystal structures. The linkers connecting the NTD and CB domains of Int adopt distinct and partially extended conformations, a situation quite different from the compact and symmetric arrangement of linkers observed in the crystal structures of Int bound to short oligonucleotide DNAs (*Biswas et al., 2005*). In addition, the central base-pairs of the HJ DNA were not well-defined in the Int-HJ crystal structure. To obtain low-energy conformations for these regions within the EM envelope and to regularize the stereochemistry of the DNA splice junctions and sites of bending, we performed molecular dynamics refinement using dynamic elastic network (DEN) restraints (*Schröder et al., 2007*). The protein•DNA sub-complexes derived from crystal structures were tightly restrained to adopt their experimental conformations, whereas the Int NTD-CB linkers and central HJ DNA base-pairs were allowed flexibility. To restrain flexible regions of the model to remain within the EM envelope, we included an energy term for agreement with the Fourier coefficients of the EM density. As shown in *Figure 5*, the final structure agrees well with the EM envelope. There are no major features lacking density and only one small region of density that was not fit by the model (discussed below). The P and P' arm trajectories are particularly well defined by the EM envelope (*Figure 5A*), as is the core Int-HJ tetramer (*Figure 5B*).

The limited resolution of the cryo-EM map does not permit construction of an atomic model with accurate detail at the level of side chains and nucleotides. Consequently, the model coordinates deposited for the excisive HJ complex specify only the positions of protein domains and the paths of the DNA duplex arms. Inter-domain linkers for the Int-C', Int-B, and Int-B' subunits are provided in arbitrary conformations in the final model to improve visualization of the structure and to facilitate modeling in future experiments; no attempt was made to predict the structures of these linkers.

## Overall structure of the λ excisive recombination intermediate

Three views of the excisive complex structure are shown in *Figure 6* and a video illustrating several views of the complex is available as *Video 1*. At the center of the excisive complex an integrase tetramer is bound to a core-site HJ, where strand exchange has taken place between the B half-site of

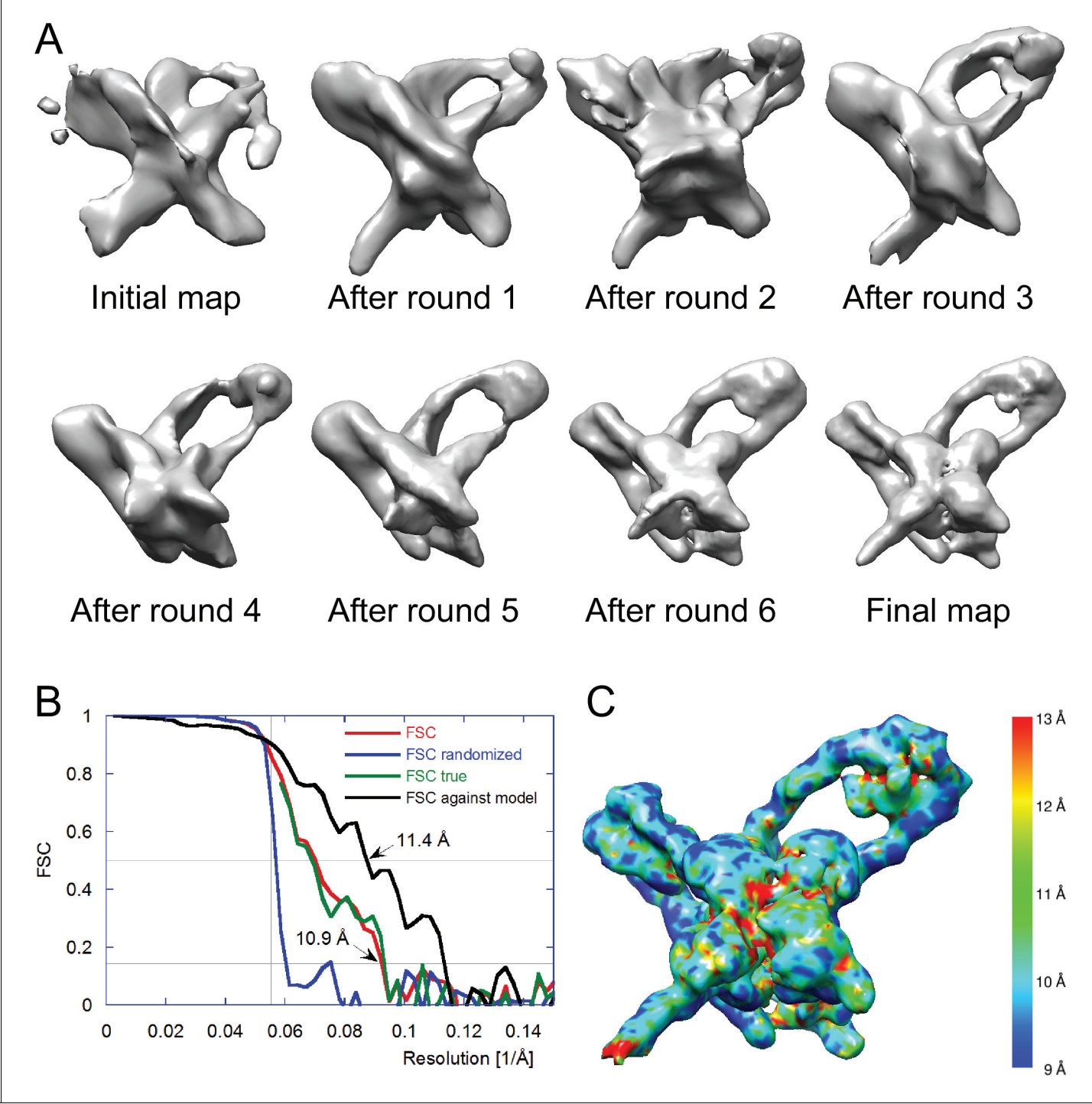

**Figure 4.** Electron cryo-microscopy refinement and 3D classification. (**A**) Different stages of refinement and classification using FREALIGN (*Lyumkis et al., 2013*). Reconstructions at each stage are shown. The initial map was generated from 2D class averages (*Figure 3C*) using EMAN2's initial model procedure (*Tang et al., 2007*). Subsequent maps resulted from several rounds of refinement and 3D classification in FREALIGN. The best-scoring reconstruction from each round was used to initialize the next round. The final reconstruction contained 10,956 particles. (**B**) Resolution estimation of the final reconstruction using the Fourier Shell Correlation (FSC) method. The red curve (labeled FSC) was calculated using the two reconstructions obtained from half the data each and masked using a tight mask that left a margin around the reconstructed density of about 15 Å. The blue curve (FSC randomized) was calculated using a second set of reconstructions obtained from data phase-randomized beyond a resolution of 18 Å. This curve was used to correct the first curve for any masking effects (*Chen et al., 2013*), yielding the final green curve (FSC true). The black curve (FSC against model) was calculated between the final map and a map derived from the atomic model presented in this study. The vertical line indicates 18 Å

*Figure 4 continued on next page*

*Figure 4 continued*

resolution, the limit beyond which no data was used for refinement and classification. The horizontal lines indicate values of 0.143, the FSC threshold used to assess the resolution of the map (*Rosenthal and Henderson, 2003*), and 0.5, the corresponding FSC threshold when comparing an experimental map against a noise-free model. Both 'FSC true' and 'FSC against model' suggest a resolution of about 11 Å. The FSC data are shown in *Figure 4—source data 1* and *2*. (C) Local resolution map calculated using ResMap (*Kucukelbir et al., 2014*). The resolution is fairly uniformly indicated as 11 Å, except for small regions within the Int tetramer and at one of the IHF heterodimer sites, indicating lower resolution and higher structural variability in these regions.

The following source data is available for figure 4:

**Source data 1.** FSC data in Excel format.
**Source data 2.** FSC data in csv format.

*att*L and the C half-site of *att*R (see *Figures 1B* and *2C*). We refer to the integrase subunits by the core half-sites to which they are bound. Int-C, for example, is the integrase subunit bound to the C half-site. The most striking features of the complex are the tight bends of the P and P' arms as they emerge from the C and C' core half-sites. IHF heterodimers bend the P arm at the H2 site and the P' arm at the H' site, in each case re-directing the DNA back towards the Int tetramer. For P', the IHF bend is sufficient for the P'1 and P'2 arm-binding sites to be engaged by the Int subunits bound at the C' and B core sites, respectively (*Figure 6B*). These are two of the three Int bridging interactions identified using biochemical and genetic approaches (*Tong et al., 2014*).

For the P arm, the IHF-mediated bend at H2 is phased differently with respect to the core tetramer compared to the bend at H'. Rather than being directed towards the NTDs of Int, the H2 bend causes the P arm to run alongside the Int tetramer. Three Xis subunits provide a second change in direction of the P arm, directing it across the top of the Int tetramer (*Figure 6A*). As a result of the IHF and Xis-induced bends, the P2 site is positioned where it can be engaged by the Int subunit bound to the B' core site (*Figure 6A and C*). The B'-P2 bridge is the third bivalent interaction required for excisive recombination (*Figure 2C*) (*Tong et al., 2014*). Remarkably, the DNA bend observed in the Xis•DNA crystal structure (*Abbani et al., 2007*) fits well into the EM envelope, which has strong density for both the DNA and the Xis subunits. The P arm trajectory is also aided by an intrinsic A-tract sequence found in the Fis binding site. Fis was not present in the complexes analyzed by EM and as expected, there is no density corresponding to Fis at this site. We note, however that Fis can be readily accommodated by the structure (*Figure 6C*), which fits well with the proposed role for Fis as a non-essential enhancer of recombination (*Ball and Johnson, 1991a*; *1991b*; *Esposito and Gerard, 2003*; *Thompson et al., 1987a*).

The only component of the excisive HJ complex that is not accounted for by EM density is the NTD of Int-C (*Figures 2C* and *6A*). This domain has no known function in excisive recombination, since only three arm-binding sites are used in the reaction (*Numrych et al., 1990*; *Thompson et al., 1987b*; *Tong et al., 2014*). One plausible location is near the Int-NTD bound at P2, where there is a small lobe of additional density present (*Figure 5A*). It is clear, however, that the Int-C NTD is not positioned directly above its connected CB domain, since the Xis molecule bound at the X1 site occupies this space. We did not fit the Int-C NTD into the P2-associated density because the domain's orientation and position could not be clearly established.

## Agreement with FRET measurements

The EM structure described here provides an opportunity to re-examine the model and data recently reported in a FRET-based study of the λ excision HJ intermediate (*Seah et al., 2014*). In that work, 12 DNA sites were labeled with fluorophores and 28 inter-site distances were estimated using in-gel fluorescence measurements. These distances, combined with knowledge of the Int bridging interactions and crystal structures of the component sub-complexes, were used to construct a model of the excisive HJ complex. The overall architecture of our EM structure is in broad agreement with the FRET-based model. As might be expected from the small number of structural restraints available, the FRET model is largely schematic in nature. For example, the tight, parallel paths of the P and P' arms were not predicted in the FRET model, nor was the extensive Xis-Int interface that links the two

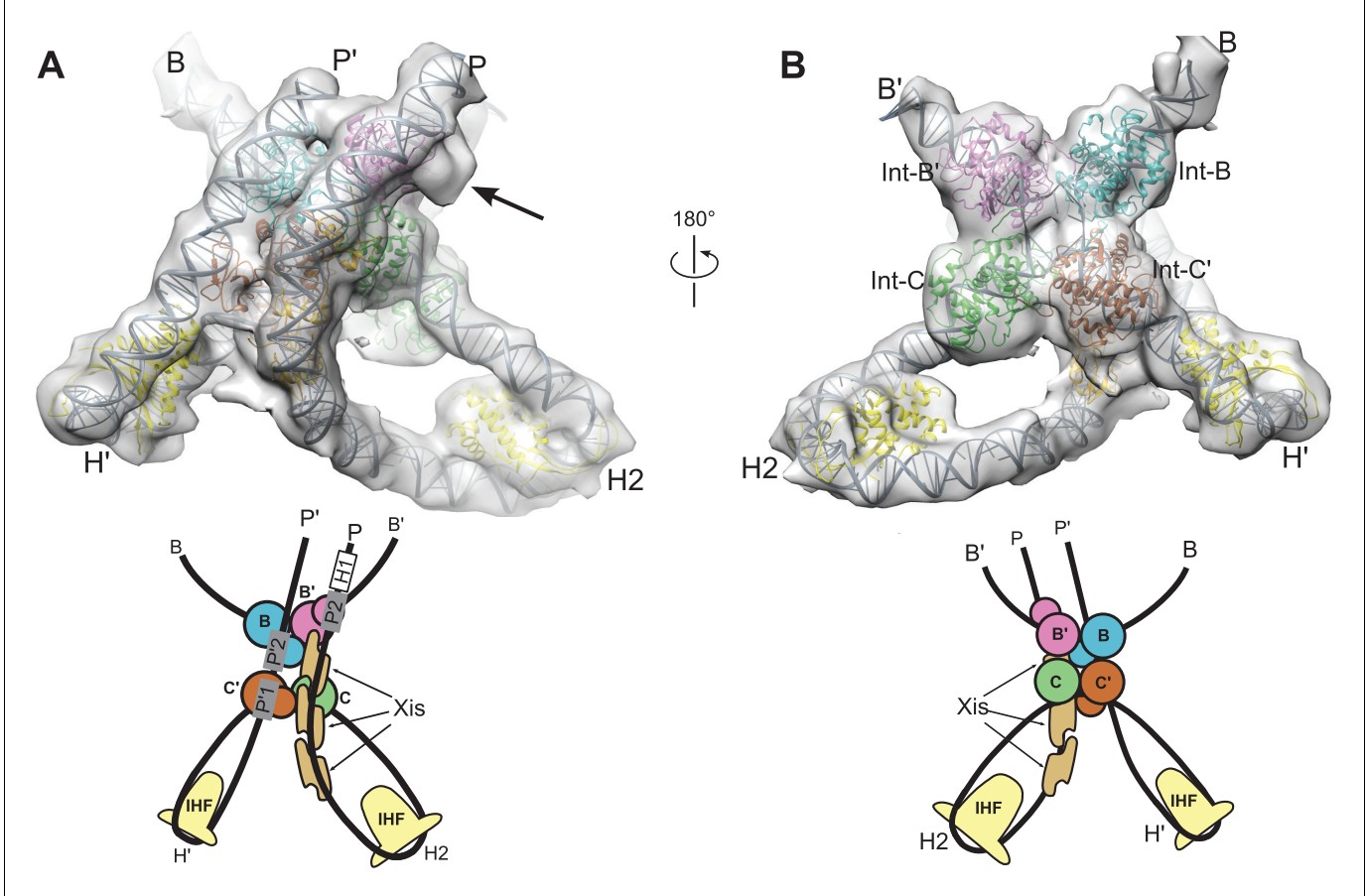

**Figure 5.** Model and reconstructed density for the λ excision complex. (**A**) View from the 'top' of the complex, illustrating the IHF-mediated bends at H2 and H' and the close, parallel trajectory of the P and P' arms. Unmodeled density that could be occupied by Int-C NTD is indicated by an arrow. (**B**) View from the bottom of the complex, highlighting the catalytic domains of the Int tetramer. The Int subunits are labeled according to the core half-sites where they are bound (**B**, **B'**, **C**, and **C'**). IHF heterodimers and Xis subunits are indicated. In the schematic representations shown below the structural models, the DNA strands near the junction centers are omitted for clarity.

arms together (see below). Indeed, the details that emerge from the EM structure provide important new insights that could not have been predicted based on distance measurements alone.

The EM structure also provides an opportunity to ask if the dye positions chosen for the FRET study were appropriate and to determine how well the EM structure agrees with the estimated distances (*Table 1*). Six of the locations appear to have been good choices, since they are largely free from steric interference that would compromise free rotation and/or the volumes accessible to the fluorophores. The EM structure is in good agreement with all of the distance estimates involving these sites. Two of the sites, *att*P- (-118) and *att*P- (+79), have serious steric conflicts based on the EM structure. The *att*P- (-118) dye on the P arm is directed into the path of the *P' arm*, possibly explaining why the arms diverge in the FRET model. The dye at *att*P- (+79) is on the P' arm and is directed into the Int-NTD bound at P2. Thus, the distances involving these sites are expected to be compromised and indeed, we see much poorer agreement with the EM structure for the distances involving these locations. The remaining four sites used for the FRET study are located on extensions of the P and P' arms that were not included in the constructs used for EM. Thus, there is excellent agreement between the EM structure and the FRET distance estimates for those dye positions that were correctly predicted to be free from steric interference.

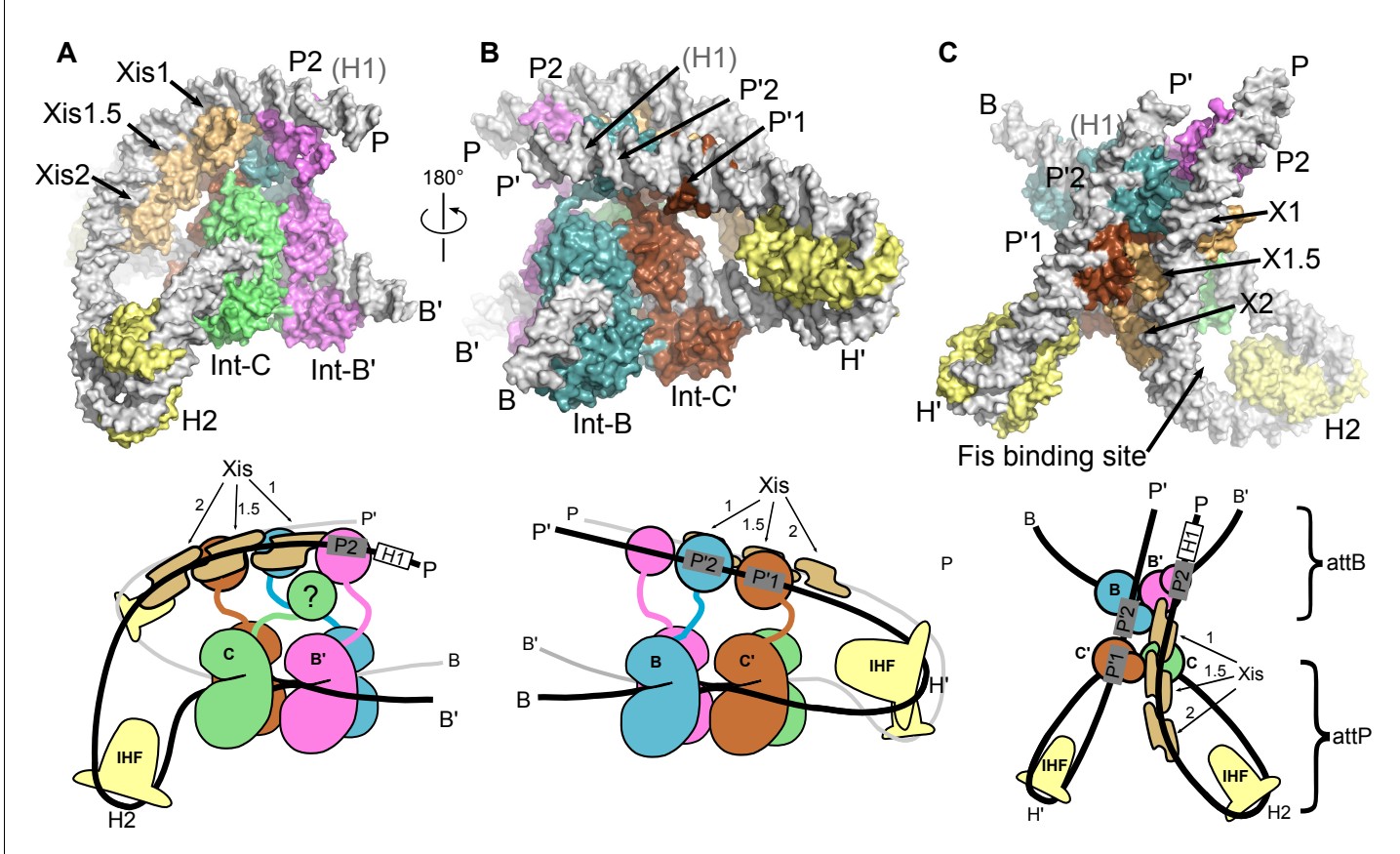

**Figure 6.** The λ excision complex structure. (**A**) View towards the *att*R-derived half of the complex, where the path of the P arm can be seen. A sharp IHF (yellow) directed bend at the H2 site, combined with the cooperative binding and bending by three Xis molecules (gold) and the Int subunit bridging between the B' and P2 sites (magenta), redirects the P arm over the top of the Int tetramer. The Int-C NTD is not present in the model because it could not be unambiguously fit into EM density (approximate location is indicated in the corresponding cartoon). **B**) View towards the *att*L-derived half of the complex, where IHF bound at the H' site sharply bends the P' arm and enables Int subunits to bridge between the C' and P'1 sites (brown) and between the B and P'2 sites (blue). (**C**) View from the top of the excision complex, where the P'-arm (from *att*L) and P-arm (from *att*R) run nearly parallel and close to one another. An intimate interface is formed between the Xis subunits and the Int-C' and Int-B NTDs. The unoccupied, but accessible binding site for Fis is indicated. In the schematic views of the excision complex shown below the structural models, the DNA strands near the junction center are omitted for clarity. For reference, the brackets labeled *att*P/*att*B indicate the relative positions of the excision products. See *Video 1* for multiple views of the excisive complex structure and isolation of individual subunits.

## An unexpected P' arm trajectory during excision

In crystal structures of Int complexes containing simplified oligonucleotide arm-binding sites, the DNA bound by Int NTDs lies parallel to the Int tetramer and the Int NTD-CB linkers form a compact network of interactions (*Biswas et al., 2005*). In contrast, the P' arm in the EM structure follows a relatively straight path after the H' bend, resulting in an upwards trajectory as it passes over the Int tetramer (*Figure 6B*). Consequently, Int-C' and Int-B linkers must adopt extended and distinct conformations in order to bridge the P' arm and the Int CB domains. Int bridge formation at P'1 and P'2 of *att*L therefore does not require bending of the P' arm after the IHF-directed U-turn at H'. Instead, Int uses the inherent flexibility present in the NTD-CB linker regions to reach the P'1 and P'2 binding sites.

## Xis protein plays multiple roles in determining directionality

A primary function of Xis is to bend the P arm to stabilize a synapsis-competent *att*R complex. The EM structure reveals that Xis also mediates a substantial protein-protein interface between the P and P' arms. The Xis molecules bound at the X1 and X1.5 sites of the P arm are packed against the

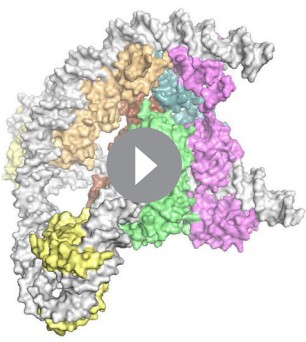

**Video 1.** The bacteriophage λ excisive HJ intermediate. Multiple views of the excisive complex are shown. The complex is stripped of all proteins to illustrate the path of the DNA arms as they loop around the 4-way junction. IHF, Xis, and Int subunits are added individually to the DNA to help visualize their structural and functional roles. The assembly sequence shown is for visualization only; nothing is implied regarding the actual order of addition or assembly during λ excision.

Int-B and Int-C' NTDs bound at the P'2 and P'1 sites, respectively (*Figure 6C* and *Video 1*). The Xis1.5-Int-C' interface appears to be particularly extensive, although at the resolution of the EM density, we cannot visualize the conformations or roles of individual side chains. These Xis-Int interfaces are presumably important, given the close approach and parallel trajectory of the P and P' arms in the excisive complex.

The identification of a Xis-mediated interface between the P and P' arms provides an explanation for how synapsis is facilitated in the excisive pathway. Int binds only weakly to the core sites of *att*L and *att*R and the Int-bound core sites do not form stable synaptic complexes that undergo recombination (*Bushman et al., 1985*). What provides the interactions that stabilize the *att*L x *att*R synaptic complex to initiate excisive recombination? Crystal structures of Int bound to core and a symmetrized mimic of arm DNA fragments suggested that the Int NTDs, along with their associated linkers, could play this role (*Biswas et al., 2005*). However, earlier experiments with a Cre-Int chimeric recombinase (*Warren et al., 2008*) and the present EM structure indicate that this is unlikely. The three Int NTDs in the excisive complex are arranged in a linear manner, where the NTD bound at P2 makes no contact with the NTDs bound at P'1 or P'2 (*Figure 6C*). Instead, Xis-NTD, and possibly Xis-linker interactions provide the additional glue required to hold *att*L and *att*R together and allow the initial strand exchange to take place.

There are 18 residues at the C-terminus of Xis that were missing in the Xis•DNA crystal structures (*Abbani et al., 2007*; *Sam et al., 2004*) and are not included in our excisive complex model. The C-termini of the Xis domains bound at the X1.5 and X2 sites extend into the gap between the P and P' arms (i.e., towards the viewer in *Figure 6C*). Six of the eighteen missing residues are arginine or lysine, suggesting that the basic C-terminal tails of Xis could interact with P and/or P' arm DNA, further stabilizing a synaptic interface. The C-terminus of Xis bound at X1 is instead directed at the Int-B NTD bound at P2, consistent with the observed cooperativity of Xis and Int binding to *att*R arm DNA (*Bushman et al., 1985*; *Numrych et al., 1992*; *Sarkar et al., 2002*). Thus, the EM structure reveals and/or explains three functional roles for Xis in excisive recombination: i) promoting Int NTD binding at the weak P2 site; ii) bending the P arm to place P2 in position for cooperative Int binding and formation of a synapsis-competent *att*R; iii) mediating an Xis•Int interface between the P and P' arms that facilitates *att*L x *att*R synapsis.

## How Fis protein functions as a gratuitous enhancer

Although Fis is not required for excisive recombination, it does stimulate the reaction when Xis is limiting (*Thompson et al., 1987a*). The views in *Figure 6A and C* provide an explanation for this stimulation. The A-tract present in the Fis site, Xis binding at X1, X1.5, and X2, and Int binding at P2 all contribute to the formation of a properly assembled P arm, and therefore a properly assembled *att*R at the start of the excision pathway. Fis binding stabilizes the A-tract bend in *att*R (*Stella et al., 2010*) and facilitates Xis binding (*Papagiannis et al., 2007*), providing additional sources of cooperativity for P arm assembly that would enhance the activity of Xis at lower concentrations.

## Minimal structural changes link the two pairs of DNA strand exchange

Structures representing reaction intermediates in the Cre and λ systems have led to a model for understanding how strand exchange is coordinated (*Aihara et al., 2003*; *Biswas et al., 2005*; *Gopaul et al., 1998*; *Guo et al., 1997*). The essential geometric features of this model for λ

**Table 1.** FRET vs. EM model distances in the λ excisive recombination complex. R(FRET) values were determined by **Seah et al. (2014)** using in-gel fluorescence measurements for dye pairs whose positions were designed in the absence of a model for the λ excision complex architecture. Dyes positioned at sites 4,5,6,7,11, and 12 are predicted to be largely free from steric interference based on the EM structure. The FRET pairs involving these positions are in the first group below. Dyes positioned at site 3 (second group) and site 8 (third group) are predicted to have steric conflicts between the dye and the excision complex. Four additional sites (1,2,9, & 10) used in the study are on extended P and P' arms not present in the construct used for EM studies. See **Seah et al. (2014)** for descriptions of the site position nomenclature.

| Position i | Position j | R (FRET) (Å) | R (FRET) Model) (Å) | R (EM Model) (Å) |
|---|---|---|---|---|
| 7 [P+50T] | 6 [P+17B] | 56 | 57 | 50.0 |
| 4 [P-58B] | 5 [P-15T] | 70 | 72 | 70.6 |
| 5 [P-15T] | 12 [B+17B] | 76 | 77 | 78.8 |
| 4 [P-58B] | 6 [P+17B] | 76 | 70 | 67.9 |
| 5 [P-15T] | 6 [P+17B] | 89 | 80 | 81.1 |
| 6 [P+17B] | 11 [B-15T] | 93 | 81 | 85.3 |
| 7 [P+50T] | 4 [P-58B] | 99 | 125 | 108.9 |
| 11 [B-15T] | 12 [B+17B] | 103 | 86 | 82.5 |
| | | | | |
| 3 [P-118T] | 12 [B+17B] | 80 | 69 | 64.1 |
| 3 [P-118T] | 5 [P-15T] | 82 | 77 | 95.3 |
| 3 [P-118T] | 11 [B-15T] | 99 | 108 | 100.0 |
| | | | | |
| 8 [P+79B] | 3 [P-118T] | 56 | 56 | 41.4 |
| 8 [P+79B] | 11 [B-15T] | 74 | 65 | 75.2 |
| 8 [P+79B] | 5 [P-15T] | 96 | 109 | 81.3 |
| 8 [P+79B] | 6 [P+17B] | 96 | 107 | 85.4 |
| 8 [P+79B] | 4 [P-58B] | 108 | 158 | 124.8 |
| 8 [P+79B] | 12 [B+17B] | 116 | 73 | 68.2 |

integrase are shown in *Figure 7*. The scissile phosphodiesters, which flank the 7-bp crossover sequence located at the center of the core sites, form a parallelogram in the HJ intermediate. The sites that are active for cleavage form the closer diagonal pair; those that are inactive are farther apart. Isomerization of the HJ intermediate between a top-strand cleavage configuration and a bottom-strand cleavage configuration involves exchanging angles within the scissile phosphate parallelogram (*Figure 7A and B*). In the Cre system, isomerization is coupled to changes in the angles between the core site duplexes that converge at the center of the junction. Changes in the interfaces between catalytic domain subunits that result from isomerization are responsible for activation/deactivation of active site pairs (*Van Duyne, 2001*).

The core HJ isomer that we fit in the EM structure corresponds to a bottom strand cleavage configuration, where the complex is poised to carry out strand exchange to form *att*P and *att*B products (*Figure 7B*). We considered how the structure of the alternative isomer, corresponding to a configuration where the top strands have just been exchanged to generate the HJ from the synapsed *att*L and *att*R partners, might differ in this complex (*Figure 7A*). Although there are significant changes near the center of the core complex where the HJ branch migrates by one bp, the ends of the duplexes remain essentially in the same position, with only small torsional changes (*Figure 7* and *Video 2*). This finding is consistent with the pseudo-fourfold symmetry of the core HJ complex (*Biswas et al., 2005*) and the local four-fold symmetry of the EM density corresponding to this region (*Figure 5B*). Thus, at the resolution of our EM data, we cannot distinguish between HJ isomers and the complexes analyzed may be a mixture of the two isomeric forms.

The nearly isosteric model for HJ isomerization shown in *Figure 7* has important implications for considering how the P and P' arms of the excisive complex might change during recombination. In

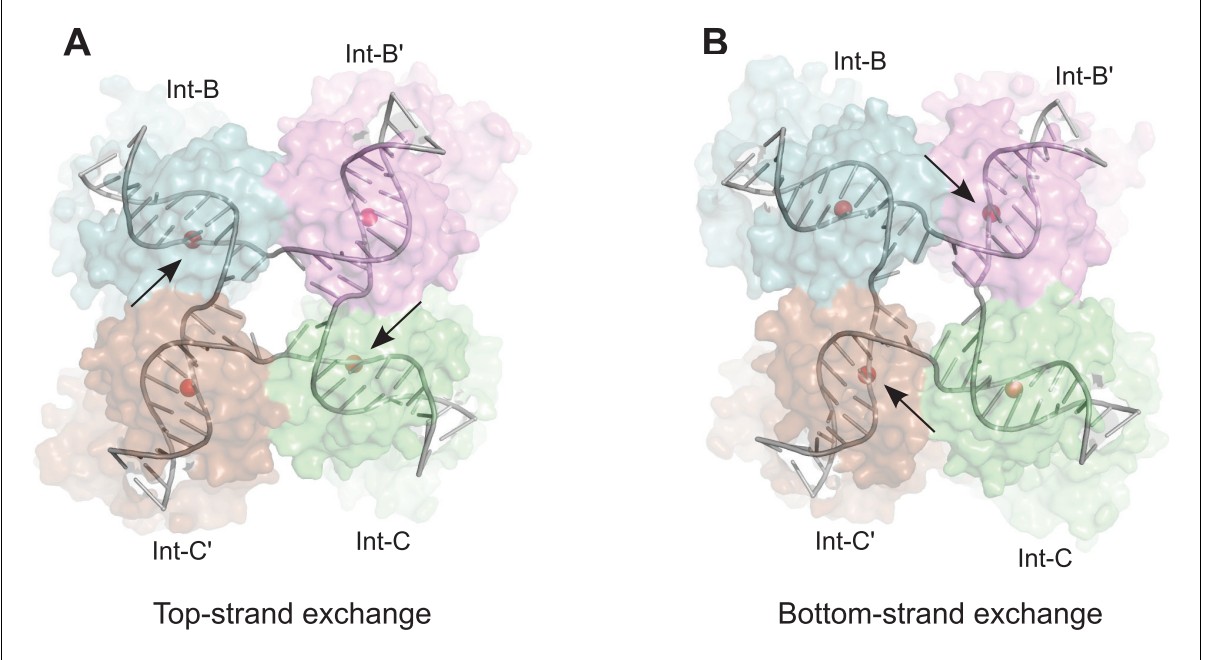

**Figure 7.** Model for isomerization of HJ intermediate. The core HJ complex is fit to the reconstructed density, based on structure 1Z1G (Protein Data Bank) (*Biswas et al., 2005*). (**A**) The HJ isomer corresponding to a top-strand cleavage configuration, where the Int subunits bound at the B and C half-sites are activated for cleavage and strand exchange. **B**) Alternative isomer, where Int-B' and Int-C' are activated for cleavage of the bottom strands. Scissile phosphates (red spheres) are closer together in the active subunits compared to the inactive ones. Arrows indicate the activated phosphodiesters. The interconversion between (**A**) and (**B**) involves migration of the branch point by one base-pair, with only minimal changes at the ends of the duplex arms. An animation that illustrates the HJ isomerization is provided as *Video 2*.

the absence of structural models, it has been difficult to rationalize how the arms could accommodate, or perhaps even direct the changes in quaternary structure expected from a Cre-like isomerization. In contrast to Cre, however, the core HJ structure observed in the EM and crystal structures of the λ intermediate indicates that isomerization could occur with relatively little impact on the flanking DNA segments. We suggest that small changes in position and torsion of the C and C' core sites can be accommodated by the in-plane and out of plane bending modes of IHF, with little impact on the Xis•Int interface assembled between the P and P' arms as they cross the Int tetramer. In other words, the EM structure implies that isomerization of the HJ intermediate can occur in the absence of large changes in quaternary structure and therefore does not require coupling to large changes in the P and P' arm positions. This type of overall isosteric isomerization may be facilitated by the odd-numbered 7-bp crossover region that is found in essentially all tyrosine integrases, e.g. (*Hakimi and Scocca, 1996*; *Kolot and Yagil, 1994*; *Peña et al., 1996*; *Williams, 2002*). A stereochemical explanation for how a one-bp branch migration accomplishes this will be an interesting topic for future studies.

## A model for the λ integration complex

The integrative HJ intermediate is formed when a complex containing *att*P DNA, an Int tetramer, and three IHF heterodimers captures the *att*B site in the host genome and carries out top strand exchange between the core DNA sequences (*Figure 1*). We previously proposed a model for this complex based on core-arm site bridge data and insights derived from the FRET-based excisive complex model discussed above (*Seah et al., 2014*). The EM structure of the excisive HJ complex provides an opportunity to revise the integrative model in the context of a well-defined molecular architecture and correctly positioned P and P' arms for the excision complex. The new integrative HJ model, constructed using a similar procedure to that previously described, is shown in *Figure 8*.

In both the integrative and excisive complexes, IHF bending at the H' site directs the P' arm over the CB domains of the Int tetramer. In the integrative complex, the B'-NTD binds at P'3, since Xis is

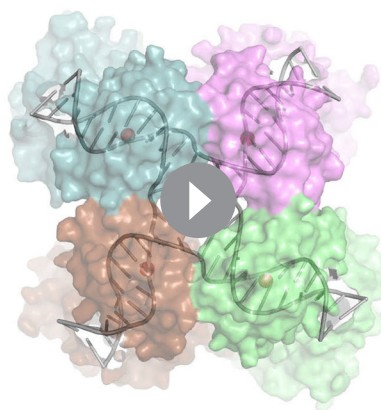

**Video 2.** Isomerization model for the λ HJ intermediate. The core HJ is shown switching between the isomers described in *Figure 7*. This type of isomerization could occur within the core of the recombination complex, without large changes in positions of the attached arms.

not present to position the P2 site to bind to this domain. A striking aspect of the new integrative complex model is that the P' arm trajectory requires no positional changes in order to readily accommodate binding of the C', C, and B' NTDs of Int (*Figure 8A*). Thus, the P' arm position and trajectory appear to be invariant in excisive and integrative recombination.

In the absence of Xis, the P arm is not directed towards the Int NTDs after the H2 bend. Instead, it follows an upward trajectory beyond the P' arm, where IHF binding at the H1 site reverses its course and directs it back towards the Int tetramer (*Figure 8B and C*). This P arm trajectory places the P1 site within reach of the Int B-NTD, a high-affinity interaction that is crucial for integrative recombination (*Tong et al., 2014*). Thus, the P arm crosses the P' arm to position Int-B during integration, but runs parallel to the P' arm and promotes Int-B' binding at P2 during excision.

The P-P' arm crossing that occurs during integration generates a negative DNA-crossing node, a feature that provides a structural explanation for the long-standing puzzle of why the topologies of integrative and excisive recombination are different and why supercoiling is required for efficient integration (*Nash, 1975*; *Richet et al., 1988*). Indeed, the integrative model shown in *Figure 8* cannot be constructed based on IHF-phased bends at H2 and H1 alone; additional DNA-bending is required, which we suggest would occur naturally in a negatively supercoiled *att*P, but would be energetically unfavorable in linear substrates. The P-P' crossing also involves close DNA-DNA contacts between the arms, explaining in part the important role of polyamines such as spermidine and/or divalent cations in the integrative reaction (*Kikuchi and Nash, 1978*; *Nash, 1975*).

## Structural basis for regulated directionality of integrative and excisive recombination

The most remarkable feature of bacteriophage λ excisive and integrative recombination is the precisely controlled unidirectionality of these two isoenergetic reactions which, at the DNA level, are the simple reverse of each other. While many features of the distinct but overlapping protein ensembles responsible for each reaction (and its regulation) have been worked out, the present study finally allows us to consolidate and incorporate these features into the structure of a fully functional reaction intermediate. As discussed above, the structure reveals an important new role for Xis, clarifies the role of the Int linker regions, defines the paths of the arms as they embrace the enzymatically active core region, and suggests that HJ isomerization occurs with only minor changes in quaternary structure. Equally important, the good fit between the present structure and the large body of genetic, biochemical, and biophysical data that exists (reviewed in [*Landy, 2015*]) makes it relatively straightforward to extend the present structure to informative models for the other intermediates involved in excisive and integrative recombination. Based on the EM structure of the excisive HJ complex and the model of the integrative HJ complex described here, our current views of the mechanisms of directionality and regulation in λ site-specific recombination are summarized below and illustrated schematically in *Figure 9*.

The biochemical feature defining excisive and integrative recombination as two different pathways (and not the forward and reverse of a single pathway) is the fact that both reactions are initiated by exchange of the same ('top') strands to form their respective HJ intermediates (*Kitts and Nash, 1988b*; *Nunes-Düby et al., 1987*). Examination of the models shows that the order of strand exchange in both pathways is determined prior to synapsis by the patterns of specific Int bridges between arm and core sites and by the IHF- and Xis-induced bends that enable those bridges. For

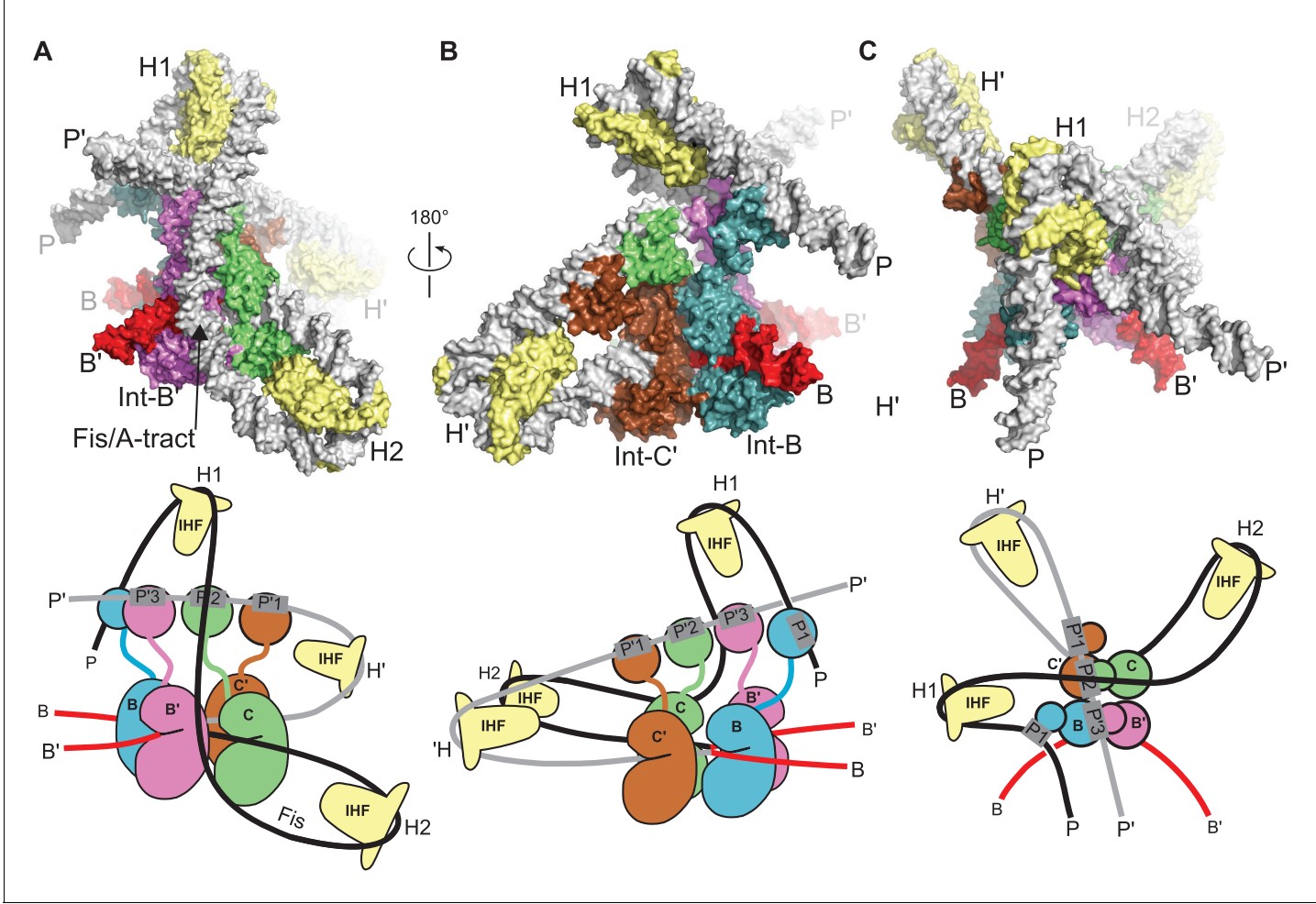

**Figure 8.** Model of the λ integration complex. (**A**) View towards the **B′**–**C** face of the complex, where the trajectory of the P arm can be seen. IHF bending at the H2 site, combined with an A-tract bend at the Fis site and additional bending provided by negative supercoiling, directs the P arm over the top of the Int tetramer and over the P′ arm. IHF-bending at the H1 site re-directs the P arm down along the opposite face of the tetramer. (**B**) View towards the **C′**–**B** face of the complex, showing Int-B bridging between the B core site and the P1 site. The catalytic and CB domains of Int-B wrap around the opposite face of the *att*B helix (with respect to the Int bound at the **B′** site). This arrangement, which is facilitated by the flexible P1 tether, must be accommodated during synapsis, as described in the text and *Figure 9*. The nearly linear arrangement of NTDs bound at P′1, P′2, and P′3 of the P′ arm can also be seen. (**C**) View from the top of the integration complex. Schematic drawings of the integrative complex are shown below the structural models, where the P arm is black and the P′ arm gray. The DNA connectivity at the HJ center is omitted for clarity; consequently the junction between the P arm (black) and P′ arm (gray) is not visible. The bend directions of the core sites are opposite to those shown in *Figure 6* for the excision complex (e.g., compare the **C**–**C′** bend in (**C**) vs. *Figure 6C*), underscoring that integration and excision are not the mechanistic reverses of one another.

example, the C′ and B bridges to the P′1 and P′2 sites of *att*L require a specific bend of the B-C′ core site, which in turn commits *att*L to top strand exchange in the synaptic complex with *att*R (*Figure 9Ea*). Similarly, formation of an integration-competent *att*P assembly requires a specific bend direction of the C-C′ core site in order to form the correct bridging interactions (*Figure 9Ia*).

We found that a stereochemically plausible *att*P model can be assembled if the C-C′ core site is bent in the opposite direction to that shown in *Figure 9Ia* (i.e., a bottom-strand cleavage configuration), but the resulting complex has a (+) crossing node and would therefore be strongly disfavored in a negatively supercoiled substrate. We conclude that only a top strand exchange integrative complex configuration is consistent with negative supercoiling.

The excisive complex structure delineates the mechanism by which Xis acts as a directionality switch between the excisive and integrative recombination pathways. During formation of a

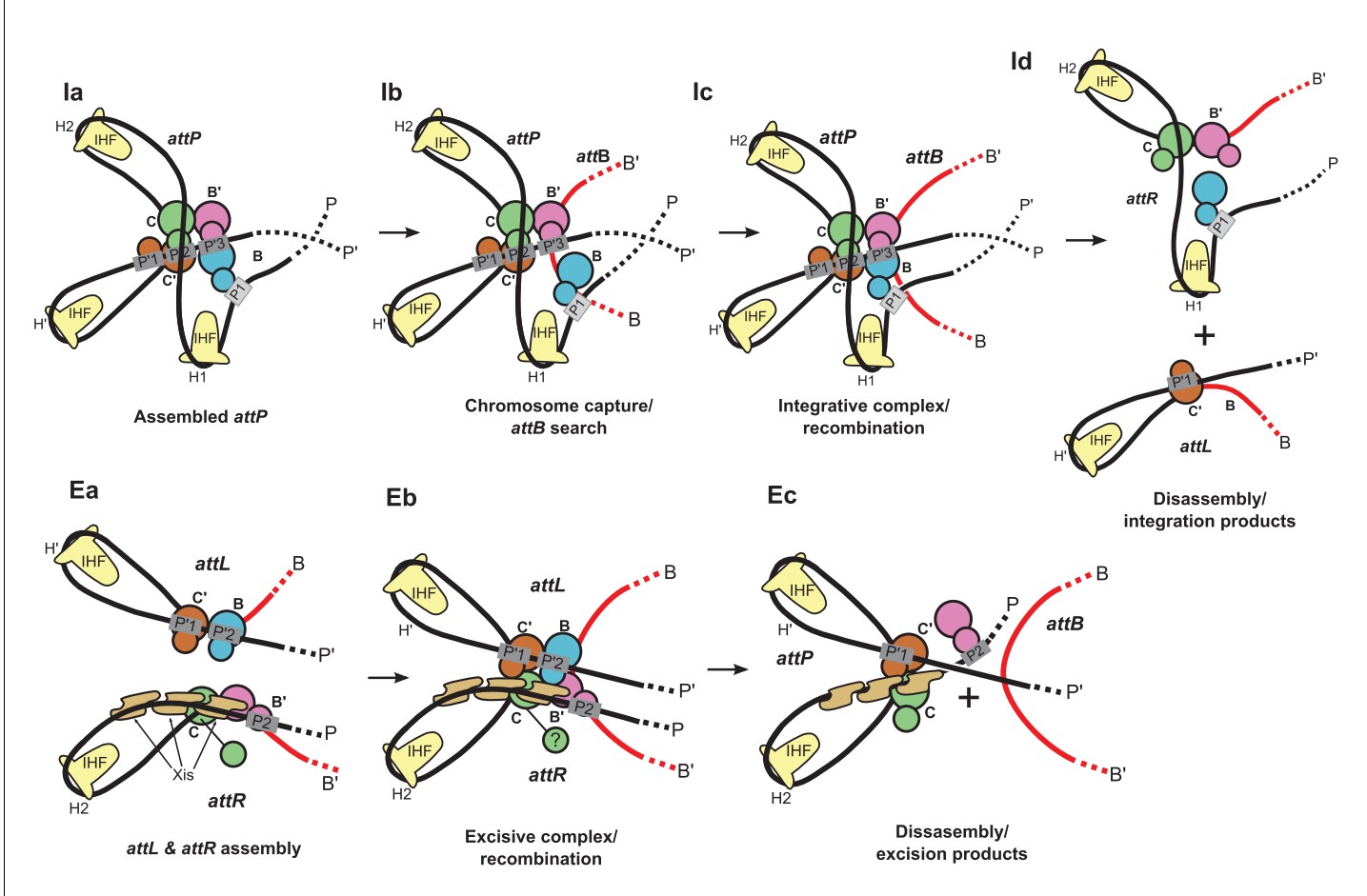

**Figure 9.** Schematic of the λ integration (**Ia–Id**) and excision (**Ea–Ec**) pathways, based on the structural models presented here. (**Ia**) In the presence of Int and IHF (but not Xis), an integration complex is assembled on supercoiled *att*P. We suggest that the Int-B CTD (blue) transiently associates with the Int-C' and Int-B' CTDs (brown & magenta) to form a tetrameric complex. (**Ib**) The P arm of the integration complex can swing open to test and engage candidate *att*B sequences and thereby deliver Int-B to the B site on the opposite face of the *att*B duplex, consistent with the implications of biochemical and genetic results (*Seah et al., 2014*; *Tong et al., 2014*) and the especially high affinity of Int NTD for P1 (*Sarkar et al., 2002*). (**Ic**) Stable *att*B binding leads to HJ formation and resolution. (**Id**) Disassembly results in unstable *att*R (no bridges) and *att*L (one bridge) complexes. (**Ea**) *Att*L forms a synapsis-competent complex with the two Int bridges, P'1-C' and P'2-B. In the presence of Xis, *att*R forms a synapsis-competent complex, due to bending of the P-arm and formation of an Int bridge at P2. Note that the C'-B and C-B' core sites are bent in the opposite directions compared to the integration products shown in (**Id**). (**Eb**) Xis mediates formation of a synaptic complex, where recombination can occur. (**Ec**) Disassembly results in an unstable *att*P complex containing only one Int bridge. Xis competes with formation of the integration complex shown in (**Ia**). In panels (**Ic**) and (**Eb**), the DNA strands near the center of the complex are omitted for clarity.

synapsis-competent *att*R complex, Xis enhances binding of the Int NTD at P2 through protein-protein interactions (*Sarkar et al., 2002*) and it enhances formation of the Int bridge between P2 and B' by pronounced bending of the P arm. Xis interactions with the Ints bound to *att*L also likely play an important role in facilitating synapsis. Thus, the Xis switch functions at multiple levels involving both intramolecular (*att*R) and intermolecular (*att*L-*att*R) stabilization of intermediates in excisive recombination (*Figure 9Ea and Eb*).

The structural models also account for why the prophage doesn't rapidly excise upon completion of integration and why the excision reaction is not run efficiently in reverse. The *att*R and *att*L products of integration have zero and one intramolecular Int bridges, respectively, and are therefore unlikely to be stable (*Figure 9Id*). Similarly, the *att*P complex formed at the end of excision has only a single Int bridge (*Figure 9Ec*). The models also explain why the integrative reaction is inhibited in the presence of Xis (*Nash, 1975*): Xis-induced bending of the P arm (*Abbani et al., 2007*) prevents

its upward trajectory in *att*P and thus defeats the proper positioning of the P1 site required for *att*B synapsis (*Figure 9Ec* vs *Figure 9ia*).

One of the especially interesting features suggested recently (*Seah et al., 2014*) and strengthened considerably by this work addresses the long-standing conundrum of how the large *att*P complex captures a naked *att*B site, whose core-type half-sites are arranged as imperfect inverted repeats (*Richet et al., 1988*). Because the openings of the two *att*P-bound Ints, positioned to bind *att*B, face in opposite directions they cannot dock *att*B by simple collision, i.e., the Int destined to bind the B core site must have the flexibility to wrap around the host chromosome from the opposite face. The integrative complex model shown in *Figure 8* has an inherently flexible P arm which we propose can transiently swing out along with Int bound at P1 (*Figure 9Ib*). This would provide the dynamic binding required to embrace the bacterial chromosome and clasp onto a synapsed *att*B (*Figure 9Ic*), a structural feature that is consistent with genetic and biochemical results (*Tong et al., 2014*) and the especially high affinity of Int NTD for the P1 site (*Sarkar et al., 2002*). The structure and models are also consistent with the changes in topology that occur when integrative and excisive recombination reactions are carried out within the same circular DNA molecule (*Crisona et al., 1999*) and with the requirement for negative supercoiling for efficient integration (*Richet et al., 1988*).

The highly asymmetric arrangement of the NTDs in the structure and models reported here contrasts sharply with the tightly packed complexes seen in crystal structures of HJ-Int tetramers bound to truncated and symmetrized arm-type 'consensus' oligonucleotides (*Biswas et al., 2005*). Indeed, the asymmetry and inherent flexibility described here are critical elements in understanding the structural basis of the recombination mechanisms; they are both consistent with, and explanatory for, all of the extensive genetic, biochemical, and topological analyses of this recombination pathway. We suggest that, in addition to their significance for understanding regulated directionality in recombination, the results described here will also serve as exemplars for thinking about the complex regulatory machines so abundant in prokaryotic and eukaryotic biology.

## Materials and methods

### Sample preparation

The HJ complex was obtained through excisive recombination of *att*R and *att*L bubble substrates using the following conditions: 55 nM *att*R, 50 nM *att*L, 30 mM KCl, 60 mM NaCl, 0.005 mg/mL BSA, 25 mM HEPES at pH 7.5, 150 nM IHF, 350 nM Xis and 350 nM Int were incubated at room temperature for 2 hr. The reaction mixture was then crosslinked with 0.0035% glutaraldehyde at room temperature for 10 min followed by quenching with 0.3 M glycine. The mixture was concentrated using Amicon ultracell 50K filters (Merck Millipore, Billerica, MA) to a volume of 100 μL and loaded onto a 2 mL sucrose gradient (22%–40% sucrose, 10 mM Tris, pH 8, 1 M Betaine) at 4°C. The gradient was run at 47,000 rpm using a TLS-55 rotor (Beckman Coulter, Inc., Brea CA) in a TL-100 tabletop ultracentrifuge (Beckman Coulter) for 16 hr at 4°C. The gradient was fractionated into 100 μL slices which were examined on a 5% native PAGE to analyze the purity of the fractions. Pure fractions that were devoid of any secondary bands or aggregations were selected. They were further concentrated using Amicon ultracell 50K filters (Merck Millipore) to a volume of 30–40 μL and were buffer-exchanged into 10 mM Tris, 50 mM NaCl buffer using prewashed Micro Biospin P-30 columns (Bio-Rad, Hercules, CA). The concentration of the buffer-exchanged sample was determined to be about 2 mg/mL by UV absorption using a Nanodrop spectrophotometer (Thermo Scientific, Wilmington, DE). The sample was either immediately frozen in liquid nitrogen for storage or used for plunging grids.

Details and data for characterizing the HJ complex have been described previously (*Tong et al. 2014*). The isolated protein-DNA HJ complex (without crosslinking) is stable in polyacrylamide gels and in solution for more than 12 hr at room temperature or 4C. Low levels of crosslinking (0.0035% glutaraldehyde) are required to stabilize the complex upon dilution.

### Electron microscopy

The purified Holliday junction complex was checked for homogeneity by screening negatively-stained samples on a Morgagni electron microscope (FEI, Hillsboro, Oregon). For cryo preparation,

we applied 3.0 µL of complex solution to a C-flat 1.2/1.3 Cu grid (400 mesh) (Protochips, Raleigh, NC), which had been glow discharged at 20 mA for 30 s. Grids were plunge-frozen with a Vitrobot Mark II (FEI) at 85% humidity, offset -2, blot time 7 s. Images were recorded on a Tecnai F30 electron microscope (FEI) operated at 300 kV and using a liquid-nitrogen cooled 626 cryo-specimen holder (Gatan Inc., Pleasanton, CA). We used the semi-automated acquisition program SerialEM (*Mastronarde, 2005*) to record 1359 movies with a Falcon II direct detector (FEI) at an underfocus set between 2.7 – 4.2 µm. Each movie consisted of 25 frames, collected in a 2 s exposure of 35.5 electrons/$Å^2$. The nominal magnification was 78,000x, corresponding to a magnification of 100,000x on the detector and 1.4 Å pixel size on the specimen. Movie frames were aligned and summed using motioncorr (*Li et al., 2013*).

## Image processing

From 1359 images we picked a total of 66,033 particles using e2boxer (*Tang et al., 2007*) from 4x binned images (pixel size on the specimen: 5.6 Å). We carried out 2D classification using the ISAC procedure (*Yang et al., 2012*) implemented in the image processing package SPARX (*Hohn et al., 2007*). Image defocus was determined with CTFFIND3 (*Mindell and Grigorieff, 2003*). We selected 52 good class averages to calculate initial maps with EMAN2 (e2initialmodel.py, [*Tang et al., 2007*]). The highest scoring map was used to initialize refinement and 3D classification in FREALIGN (*Lyumkis et al., 2013*), as described in the main text and *Figure 4A*. The resolution was limited to 18 Å throughout this refinement. The final reconstruction was calculated using 2x binned data (pixel size on the specimen: 2.8 Å) and had a resolution of about 11 Å (*Figure 4B*), as indicated by the FSC = 0.143 threshold criterion (*Rosenthal and Henderson, 2003*). For this estimate, the two 3D reconstructions obtained from half the data and used to calculate the FSC curve were masked with a tight mask that added a margin of about 15 Å around the reconstructed density of the Holliday junction complex. To assess how the mask affected the FSC curve, the 3D reconstructions were recalculated with particle images with randomized phases beyond 18 Å resolution and used to correct the FSC curve obtained from the masked reconstructions (*Chen et al., 2013*). The resolution of about 11 Å was corroborated by calculating the FSC between the map and atomic model presented in this study (*Figure 4B* ), and by the local resolution map (*Kucukelbir et al., 2014*); *Figure 4C*. The final map was filtered at 11 Å resolution and sharpened with a B-factor of -2500 $Å^2$. Source data for the FSC curves shown in *Figure 4B* are given in *Figure 4—source data 1* and *Figure 4—source data 2*.

## Construction and fitting of the excisive HJ complex

Construction of an excisive HJ complex model began with a core complex derived from the 3.8 Å crystal structure of full-length Int bound to unlinked HJ and arm DNA fragments (*Biswas et al., 2005*). The core complex consists of the catalytic and CB domains of Int (residues 75–356) and a 4-way DNA junction containing the C, C', B, and B' core half-sites. A P' arm complex containing IHF bound at H' and Int NTDs (residues 1–55) bound at P'1 and P'2 was constructed essentially as described (*Seah et al., 2014*), except that the NTD•DNA segments were derived from the NMR structure of an NTD•P'2 DNA complex (*Fadeev et al., 2009*). A P arm complex was also constructed as previously described (*Seah et al., 2014*), except for the use of the NMR-based model for the NTD•P2 segment. The assembled P' arm was joined to the C' half-site and the P arm was joined to the C half-site to generate the initial excisive complex model. During all construction steps, DNA segments were joined by superimposing the flanking duplexes on an idealized B-DNA splint fragment, ensuring smooth continuity of DNA twist. DNA segments were extended by superposition of idealized B-DNA fragments of the appropriate length, using a similar overlap.

The initial complex was fit manually into EM density as a rigid body. Excellent agreement with the core complex was readily obtained, but the P and P' arms were not well-positioned in the envelope. The P and P' arms were fit into density by iterative testing of small ( ± 1°) roll angles in the protein-free regions and by varying the IHF-mediated DNA bends. Changes in bending and torsion of IHF•DNA complexes (<10°) were made by rotating the DNA duplexes about the existing kinks (in-plane) or rotating the DNA arms about an axis connecting the two kink positions (torsion). The phosphate backbones of spliced, extended, and bent DNA segments were regularized to correct small deformations by highly restrained refinement in CNS (*Brunger, 2007*). To complete the model, the NTD-CB linker segments (residues 56–74) for the C', B, and B' subunits were fit into the EM

envelope with essentially arbitrary conformations. The final model was refined with CNS using dynamic elastic network (DEN) restraints and torsion angle dynamics to obtain reasonable linker geometries and good overall stereochemistry (*Schröder et al., 2007*; *2010*). To restrain flexible regions of the model to remain within the EM envelope and to optimize the final model fit to density, we included an energy term for agreement with the Fourier coefficients of the EM density.

### Construction of an integrative recombination model

An integrative HJ model was built from the excisive HJ complex based on the different bridging interactions that form during integrative recombination (*Figure 2B*; *Tong et al., 2014*). Binding of the C', C, and B' NTDs to the P' arm could be readily accommodated by changing the NTD-CB linker conformations; no change in the P' arm was required. A new P arm was assembled in the absence of Xis as previously described (*Seah et al., 2014*), using the NMR-based model for Int B-NTD binding to the P1 site. The initial model, lacking the phased bend induced by Xis, clearly indicated that the P arm must pass over P', with IHF bending at the H1 site responsible for re-directing the P1 site towards the Int tetramer. Small roll angle changes and bending at H1 and H2 (< 10°) were used to obtain a P arm geometry that avoids steric clashes and positions the Int NTD bound at P1 close to the Int B CB domain. A small amount of bending of the P' site was found to better accommodate the P arm crossing. After insertion of CB-NTD linkers for each of the Int subunits (arbitrary conformations), the model was optimized using CNS to obtain reasonable linker geometries and good stereochemistry elsewhere.

### Structural representations

*Figures 3–5* were generated with UCSF Chimera (*Pettersen et al., 2004*). *Figures 6–8* were generated with Pymol (*Delano, 2002*).

## Acknowledgements

We thank Mike Rigney for help with electron microscopy, and Christine Lank, Joanne Nelson, and Joan Boyles for help with complex preparation. This work was supported by NIH grants GM0627 and GM033928 (AL), GM108751 (GV), and P01 GM62580 (NG).

## Additional information

### Competing interests

NG: Reviewing editor, *eLife*. The other authors declare that no competing interests exist.

### Funding

| Funder | Grant reference number | Author |
| --- | --- | --- |
| National Institute of General Medical Sciences | GM0627 | Arthur Landy |
| Howard Hughes Medical Institute | | Nikolaus Grigorieff |
| National Institute of General Medical Sciences | GM033928 | Arthur Landy |
| National Institute of General Medical Sciences | GM108751 | Gregory D Van Duyne |
| National Institute of General Medical Sciences | GM62580 | Nikolaus Grigorieff |

The funders had no role in study design, data collection and interpretation, or the decision to submit the work for publication.

### Author contributions

GL, initiated the project, purified and characterized the complexes, prepared specimens for EM, collected EM data, performed particle picking, processed the EM data, contributed to the writing of

the manuscript; CX, AFB, prepared specimens for EM and collected EM data, contributed to the writing of the manuscript; DW, NS, WT, purified and characterized the complexes, contributed to the writing of the manuscript; LS, purified and characterized the complexes, performed particle picking, contributed to the writing of the manuscript; NG, initiated the project, processed the EM data, supervised the research, contributed to the writing of the manuscript; AL, GDVD, initiated the project, interpreted the EM density and built the atomic model, supervised the research, contributed to the writing of the manuscript

### Author ORCIDs
Nikolaus Grigorieff, http://orcid.org/0000-0002-1506-909X
Gregory D Van Duyne, http://orcid.org/0000-0003-0247-1626

## Additional files

### Major datasets

The following datasets were generated:

| Author(s) | Year | Dataset title | Dataset URL | Database, license, and accessibility information |
|---|---|---|---|---|
| Grigorieff N, Van Duyne G, Landy A | 2016 | Lambda excision HJ intermediate | https://www.ebi.ac.uk/pdbe/entry/emdb/EMD-3400 | Publicly available at Protein Data Bank in Europe (accession no: EMD-3400) |
| Van Duyne G, Grigorieff N, Landy A | 2016 | Lambda excision HJ intermediate | http://www.rcsb.org/pdb/search/structid-Search.do?structureId=5J0N | Publicly available at Protein Data Bank (accession no: 5J0N) |

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
