## [Decision Letter]

Thank you for submitting your work entitled "Structure of a Holliday Junction Complex Reveals Mechanisms Governing a Highly Regulated DNA Transaction" for consideration by *eLife*. Your article has been reviewed by two peer reviewers, and the evaluation has been overseen by Sjors Scheres as the Reviewing Editor and John Kuriyan as the Senior Editor.

The following individuals involved in review of your submission have agreed to reveal their identity: Andres Leschziner (peer reviewer) and Reid Johnson (peer reviewer).

The reviewers have discussed the reviews with one another and the Reviewing Editor has drafted this decision to help you prepare a revised submission.

Summary:

The excisive intasome structure of the λ phage presented here represents the culmination of decades of biochemistry and structural biology (including some misleading X-ray structures) into an atomic model based on an 11 Å cryo-EM envelope. Lambda phage is one of the classic systems for understanding site-specific recombination-its enzymology, regulation and topology. Many structures have been reported for components of the system, as well as complexes between the integrase (Int) and shorter DNA fragments, but this is the first structure of the entire assembly and, as such, is a landmark for the field. This structure will be of interest to a broad audience, as generations of biologists grew up on the molecular biology of lambda, and one of the reviewers referred to the structure as an "Ah moment" for those in the site-specific recombination-transposition field. Therefore, there was general agreement that this paper would make an excellent contribution to *eLife*, and the suggestions for improvement below are all relatively minor.

Suggestions for revision:

As you can read in the detailed comments from the two original reviews that are copied for your information below, one of the reviewers wonders whether any previous data exists to address the additional postulated roles of Xis in promoting excision. This was further discussed online, and the conclusion was that it did not seem that the authors could clearly address their postulate with past data and new experiments to test this would be difficult. There was also a question regarding data behind the revision in the location of the P-P′ crossing in the integrative intasome model. However, in principal, none of the reviewers were against including this speculative model in the manuscript.

Reviewer #1 raises questions about the plausibility of the modeled conformation of the DNA, in particular within the IHF interfaces. Also this was discussed online, and the conclusion was that most likely this was caused by the relatively low-resolution of the map combined with either the manual building or the molecular dynamics procedure that was used to refine the final model. Given the relatively low-resolution of the map, reliable atomic model building is expected to be very difficult. Perhaps the authors could include a cautionary note in their revision, where the perhaps less structurally-inclined reader is warned against interpretations of the atomic model in too much detail? In addition, the limited resolution of the reconstruction probably stems from structural variability in the sample, probably in particular from flexibility in the DNA, so that the proposed model would at best be one out of many possible structures. Therefore, the addition of a local resolution map to inform about the most flexible parts of the structure was deemed informative.

In the online discussion, there were further comments on a perceived lack of clarity with which the cryo-EM image processing was described. So please take note of the suggestions by reviewer #2 to improve this. In addition, it would be insightful to include an FSC curve between the atomic model and the EM map.

Finally, reviewer #2 suggests improvements to some of the figures. Please take particular note of the suggestions to improve consistency of Figure 9 and the chirality of the IHF arms in the model figures. Regarding the suggested changes to the movie and the cartoon: clearer figures and movies may indeed help to better convey the complicated organisation of your structure, but the extent to which you choose to modify the existing ones is left to your own judgement.

Detailed points by both reviewers:

*Reviewer #1:*

Arguably, the most significant new aspect of the present cryo-EM based model is the close association of P and P' arm segments, suggesting an attractive additional function of Xis involving trans interactions with the P′ arm. (Note that in the fourth paragraph of the subsection “Structural basis for regulated directionality of integrative and excisive recombination”, the sentence should be modified to say "likely" or "are postulated to" play an important role in facilitating synapsis). The authors discuss many other features of the model that explain biochemical properties of the reaction, particularly with respect to its directional regulation.

A more speculative model for the integrative intasome assembly is also presented that somewhat modifies the 2014 Seah et al. model, particularly with respect to the location of the P-P′ arm crossing. It is not clear to me what data is driving the revision. Although I don't believe we have learned that much new from the revised integrative intasome model, I am not opposed to including it here for the sake of completeness.

Specific comments:

There appears to be considerable modeling "freedom" in changing the path of DNA around the IHF dimers (especially H2) that seems to be necessary to generate the required writhe. The DNA paths deviate considerably from the 1IHF X-ray structure where the DNA is curved in a planar fashion as it follows along the basic sides of the protein. Is the DNA conformation within the IHF interfaces in the model structurally plausible?

In the subsection “Minimal structural changes link the two pairs of DNA strand exchange “– I am unclear on how the alternative HJ isomer (Figure 7) was modeled – what is the alternative HJ isomer structure specifically based on?

In the second paragraph of the subsection “Electron cryo-microscopy structure determination “– What is meant by 'randomly' assigned into 6 classes?

What is being referred to in the last sentence of the Abstract: "…many excisive and integrative recombination pathways, and they help explain their regulation"? I don't believe the various alternative pathways proposed for λ recombination are discussed (and don't need to be) here.

In the subsection heading 'An unexpected P' arm trajectory during excision', what is so unexpected? It seems very similar to earlier models and predicted from straightforward modeling without additional DNA bending-writhing.

In the sixth paragraph of the subsection “Structural basis for regulated directionality of integrative and excisive recombination”, what are the openings? The re-positioning of the Int-B subunit associated with the P1 site into a configuration competent to capture a naked attB seems like a very difficult step.

Would a color map displaying local resolution limits of the model be informative? Are the DNA loops as rigid as implied from the density maps?

*Reviewer #2:*

I have listed a few minor scientific comments below. My main concerns relate to how material is presented in the manuscript. Recombination is a rather impenetrable field for the non-cognoscenti and I feel the authors could have done a better job making their results accessible to the widest possible audience. I have a section below where I expand on these concerns.

Scientific points:

The authors speculate on the role of Xis in promoting recombination. They suggest that interactions between Xis and the Int-NTD stabilize the attL-attR complex. Is there any data supporting the role of this interaction? Are there known mutants in Xis that abolish or impair recombination whose general location agrees with the authors' model for their location in the excisive complex?

The authors also discuss the possible interaction between the highly basic (Arg, Lys) C-terminal tails of Xis and P/P' arm DNA and how this interaction could also stabilize the synapse. Have they (or anyone else) mutated these residues to test this idea?

The description of the EM data processing ("Electron cryo-microscopy structure determination") will not be easy to follow for non-experts. The distinction between 2D and 3D class averages is not made very clear in the text and people with passing familiarity with EM data processing will likely think that "particle alignment and classification" refers to 2D processing. Also, the 3D reconstructions generated during the 3D classification processing are called "classes" in the text but "reconstructions" in Figure 4. I think both are OK as long as they are used consistently. I would still recommend adding 2D and 3D when necessary to draw distinctions.

Presentation points:

Video 1: This movie has the potential for being the centerpiece of this paper yet all it does is show the structure rotating around a single axis. We don't even get to see the Holliday junction down its pseudo four-fold axis. Given how complicated the structure is and that many readers (particularly those not familiar with the λ system or not that structurally inclined) will have a hard time translating the cartoons shown in the figures into 3D structures, this movie could be the one point where all these complexities can be made accessible. If nothing else, the structure should rotate around both the vertical and horizontal axes. But why not go further? After introducing the entire structure, the authors could strip all the proteins out to show the path of the DNA, then add each protein component one at a time, then remove it and go to the next. (For example, show the IHFs, remove them, show the different Int monomers, remove them, show the Xis's, etc. Proteins could also be shown in pairs to highlight their interactions.) It would further help if the structure rotated while those components are displayed. The movie could end by slowly adding, and keeping, the protein components, to build back the entire complex. It is true that readers could download the structure and do this themselves but the reality is that only people with a direct interest in λ recombination would do it.

Figure 9: This figure would benefit from a little better labeling of the protein components. The arrows inside the B/C/B'/C' boxes in panel A are very hard to see. I think the Integrative Holliday junction should be shown with as much detail as the Excisive one given that an important part of the paper is modeling the Integrative complex. It would be very useful for the reader to see the different location of the Int bridges in the two complexes. I found the color scheme in panel B confusing. Why do the Int ovals change colors between the top and middle cartoons?

Figure 5: Although I realize this would take some work, a more geometrically realistic cartoon would help readers orient themselves better when looking at the structure. (For example, the IHF bound to H' is shown face-on in the cartoon while in the structure is appears edge-on, with the DNA in front.) Also, the handedness of the IHF is wrong in the cartoon in panel B (i.e. the wrong IHF arm is in front of the DNA). It would be useful if the putative density for the NTD of Int-C were indicated in this figure.

Figure 6: It would help to add the axis of rotation relating the cartoons in A and B. It may not be obvious to everyone that the axis is perpendicular to the page, rather than parallel to it as it is the case for the structures. Also, the legend should specify that the structure in C corresponds to the cartoon viewed from the front.

Figure 8: Every IHF in this figure has the wrong handedness (i.e. the wrong arm is in front of the DNA).

---

## [Author Response]

Reviewer #1:

*Arguably, the most significant new aspect of the present cryo-EM based model is the close association of P and P' arm segments, suggesting an attractive additional function of Xis involving trans interactions with the P′ arm. (Note that in the fourth paragraph of the subsection “Structural basis for regulated directionality of integrative and excisive recombination”, the sentence should be modified to say "likely" or "are postulated to" play an important role in facilitating synapsis). The authors discuss many other features of the model that explain biochemical properties of the reaction, particularly with respect to its directional regulation.*

We have modified that sentences as suggested.

*A more speculative model for the integrative intasome assembly is also presented that somewhat modifies the 2014 Seah et al. model, particularly with respect to the location of the P-P′ arm crossing. It is not clear to me what data is driving the revision. Although I don't believe we have learned that much new from the revised integrative intasome model, I am not opposed to including it here for the sake of completeness.*

The primary motivation for constructing a revised integrative recombination model is that we now have an experimental model for the excision complex to use as guide. Since the IHF bends at H' and H2 are crucial to establishing the initial trajectory of the P and P' arms, we felt that an updated model would benefit from knowing what those trajectories actually are (at the resolution of the model, of course). The result is that we are very confident of the P' arm trajectory and the general path of the P arm trajectory. There is obviously a lot of flexibility in the P arm, given the long stretch of naked DNA between H2 and H1, so the conformation we have modeled as it crosses over the complex is one of many possibilities that all look quite similar and lead to the same conclusions.

Specific comments:

There appears to be considerable modeling "freedom" in changing the path of DNA around the IHF dimers (especially H2) that seems to be necessary to generate the required writhe. The DNA paths deviate considerably from the 1IHF X-ray structure where the DNA is curved in a planar fashion as it follows along the basic sides of the protein. Is the DNA conformation within the IHF interfaces in the model structurally plausible?

The DNA paths do not deviate considerably from the IHF-DNA crystal structure. We have intentionally restricted the IHF bending angles to prevent this. It is likely that the reviewer is seeing the bending that occurs outside of the H2 site, where the DNA has indeed been bent to follow the clear path outlined by the EM envelope. To measure the out-of-plane bending of DNA in the IHF-DNA modules, we calculate the A-B-C-D dihedral angle where A, B, C, and D refer to the centers of the i-15, i-3, i+3, i+15 base pairs, respectively, and the base pair centers are the midpoints of C1' atoms. The index i refers to the central base-pair in the site. For the IHF-DNA crystal structure, this angle is -30 degrees, corresponding to bending of each arm ~15 degrees out of the best plane through the DNA. This is similar to the small out of plane bending noted by Phoebe and Howard in their paper, where they pointed out that the sign of the angle is appropriate for negative supercoiled DNA. In the modeled H2 bend, the angle is -39 degrees. Thus, we have increased the out-of-plane component by ~4.5 degrees for each arm. For the H' bend, the angle is -25 degrees, where we have decreased each arm by ~2.5 degrees. These small changes appear to be easily accommodated by the IHF heterodimer. At 11 Å resolution, we of course cannot visualize what types of corresponding small changes might occur in the IHF protein in response.

In the subsection “Minimal structural changes link the two pairs of DNA strand exchange “– I am unclear on how the alternative HJ isomer (Figure 7) was modeled – what is the alternative HJ isomer structure specifically based on?

We assumed here that HJ isomerization is symmetric. This is the same assumption that was made for Cre recombinase isomerization. To generate the alternative isomer, the structure is rotated 90 degrees about the pseudo-fourfold symmetry of the HJ intermediate, and the protein subunits and DNA strands are then re-labeled accordingly. Top strand conformations become bottom strand conformations, active subunits become inactive subunits, etc. The finding that this could all be done within the central (core) confines of the excision complex with only minor adjustments required to connect to the fixed arms was what convinced us that the arms don't need to be active participants in the isomerization process that links the strand exchange steps during recombination.

*In the second paragraph of the subsection “Electron cryo-microscopy structure determination “– What is meant by 'randomly' assigned into 6 classes?*

The goal of classification is to assign particles with common structural features to a class. If successful, different classes will represent different structures and reconstructions calculated from particles assigned to the same class will show distinctive features. If the assignment is not known a priori (as is the case here), particles are randomly assigned to different classes and, using an iterative procedure, these assignments are improved (refined) until convergence is reached, i.e. there is no further change in the assignments. To clarify the meaning of ‘randomly assigned’ the text was reworded to “Starting with reconstructions calculated from six randomly selected subsets of the data, particles were classified into six classes and their alignments refined in 40 iterations in which the resolution limit was increased in regular steps to a final limit of 20 Å.”

What is being referred to in the last sentence of the Abstract: "…many excisive and integrative recombination pathways, and they help explain their regulation"? I don't believe the various alternative pathways proposed for λ recombination are discussed (and don't need to be) here.

We were referring to the alternative reactions that λ int can catalyze, such as the 'bent-L pathway', but we agree that this is somewhat vague and is unlikely to be of value to the general readership. We have re-worded this.

In the subsection heading 'An unexpected P' arm trajectory during excision', what is so unexpected? It seems very similar to earlier models and predicted from straightforward modeling without additional DNA bending-writhing.

It was unexpected because it is relatively straight as it leaves the H' bend. This means that the P' arm cannot 'sit flat' on the Int tetramer as occurs in the Biswas et al. structure and model and in our previous FRET-based model. Additional bending is required to accomplish this. We agree that if one ignores previous models and structures, then the simplest thing is to just bend at H' and let the Int subunits bound at C' and B interact with unique linker conformations. But, to our knowledge, no one actually modeled that or suggested that it would occur like this.

In the sixth paragraph of the subsection “Structural basis for regulated directionality of integrative and excisive recombination”, what are the openings? The re-positioning of the Int-B subunit associated with the P1 site into a configuration competent to capture a naked attB seems like a very difficult step.

Agreed. But somehow, it has to happen. The NTD-P1 interaction is very tight. We don't know how tightly the Int catalytic domain interacts with the B' and C' catalytic domains in the absence of DNA; this could be quite weak. In this sense, the model we have proposed may be overemphasizing this interaction. In other words, the Int-B cat+cb domains may be so loosely associated that it is easier than it appears. Similarly, the Int-B' subunit could be loosely bound, so both subunits can search/wrap in a coordinated manner. We have also added a reference to the fact that Int binds the P1 site with higher affinity than all of the other arm sites, consistent with the suggested model for synapsis. We agree that this aspect of the model is speculative, but feel that the general nature of how integration must work in the context of a pre-formed attP complex is an important insight.

*Would a color map displaying local resolution limits of the model be informative? Are the DNA loops as rigid as implied from the density maps?*

We have calculated a local resolution map, which is now included in the Figure 5, panel C. It shows that the resolution is fairly uniform. Nevertheless, the region with the lowest resolution occurs within the Int tetramer at the center of the structure. Presumably, this reflects some variability in this region, perhaps due to conformational variability. The local resolution map implies that the DNA loops are not significantly more flexible than other parts of the structure, as the reviewer suggests. A description of the local resolution map and its interpretation has been included in the figure legend.

Reviewer #2:

*I have listed a few minor scientific comments below. My main concerns relate to how material is presented in the manuscript. Recombination is a rather impenetrable field for the non-cognoscenti and I feel the authors could have done a better job making their results accessible to the widest possible audience. I have a section below where I expand on these concerns.*

Scientific points:

The authors speculate on the role of Xis in promoting recombination. They suggest that interactions between Xis and the Int-NTD stabilize the attL-attR complex. Is there any data supporting the role of this interaction? Are there known mutants in Xis that abolish or impair recombination whose general location agrees with the authors' model for their location in the excisive complex?

We looked for Xis mutants in the literature as soon as we realized what Xis was doing in the excision complex. Most of the focus on Xis has been on DNA binding/bending and cooperative binding of Int at P2. We were unable to find studies where other Xis residues had been targeted or characterized from screens. Our new excision complex structure paves the way for structure-function studies that were not possible in the past, including not only Xis, but the Int NTDs that are involved in trans NTD-arm interaction and NTD-Xis interactions.

*The authors also discuss the possible interaction between the highly basic (Arg, Lys) C-terminal tails of Xis and P/P' arm DNA and how this interaction could also stabilize the synapse. Have they (or anyone else) mutated these residues to test this idea?*

To our knowledge, no. As noted above, these would be informative new experiments that can be done now that a structure model is available.

*The description of the EM data processing ("Electron cryo-microscopy structure determination") will not be easy to follow for non-experts. The distinction between 2D and 3D class averages is not made very clear in the text and people with passing familiarity with EM data processing will likely think that "particle alignment and classification" refers to 2D processing. Also, the 3D reconstructions generated during the 3D classification processing are called "classes" in the text but "reconstructions" in Figure 4. I think both are OK as long as they are used consistently. I would still recommend adding 2D and 3D when necessary to draw distinctions.*

Agreed. All references to density maps are now called 'reconstruction' and where classification is mentioned, it is clearly indicated whether it refers to 2D or 3D.

Presentation points:

*Video 1: This movie has the potential for being the centerpiece of this paper yet all it does is show the structure rotating around a single axis. We don't even get to see the Holliday junction down its pseudo four-fold axis. Given how complicated the structure is and that many readers (particularly those not familiar with the λ system or not that structurally inclined) will have a hard time translating the cartoons shown in the figures into 3D structures, this movie could be the one point where all these complexities can be made accessible. If nothing else, the structure should rotate around both the vertical and horizontal axes. But why not go further? After introducing the entire structure, the authors could strip all the proteins out to show the path of the DNA, then add each protein component one at a time, then remove it and go to the next. (For example, show the IHFs, remove them, show the different Int monomers, remove them, show the Xis's, etc. Proteins could also be shown in pairs to highlight their interactions.) It would further help if the structure rotated while those components are displayed. The movie could end by slowly adding, and keeping, the protein components, to build back the entire complex. It is true that readers could download the structure and do this themselves but the reality is that only people with a direct interest in λ recombination would do it.*

We agree with this suggestion and have put considerable effort into improving the video. Our new video shows several views (with rocking), adds subunits one at a time, and zooms in on the Xis-NTD and NTD-DNA interfaces that glue the P and P' arms together in the complex. We have added subtitles to annotate the structural transitions. In addition to providing 3D views that illustrate the points made in the manuscript, the video provides an independent summary of what is covered in the paper. Although it was a lot of work, we thank the reviewer for 'pushing' us to improve the video.

*Figure 9: This figure would benefit from a little better labeling of the protein components. The arrows inside the B/C/B'/C' boxes in panel A are very hard to see. I think the Integrative Holliday junction should be shown with as much detail as the Excisive one given that an important part of the paper is modeling the Integrative complex. It would be very useful for the reader to see the different location of the Int bridges in the two complexes. I found the color scheme in panel B confusing. Why do the Int ovals change colors between the top and middle cartoons?*

In order to make the suggested improvements, we split Figure 9 into two new figures and re-numbered successive figures accordingly. One of the primary limitations on clarity in Figure 9 was that we were trying to put the integrative pathway, the excisive pathway, and the HJ trapping strategy all into one figure. The new Figure 9 is the introductory figure, illustrating the excisive and integrative pathways. The new Figure 1 outlines the HJ trapping strategy. In Figure 1 (new number) we have more clearly distinguished the cleaving from inactive Ints by showing a Tyr342 symbol on the former.

*Figure 5: Although I realize this would take some work, a more geometrically realistic cartoon would help readers orient themselves better when looking at the structure. (For example, the IHF bound to H' is shown face-on in the cartoon while in the structure is appears edge-on, with the DNA in front.) Also, the handedness of the IHF is wrong in the cartoon in panel B (i.e. the wrong IHF arm is in front of the DNA). It would be useful if the putative density for the NTD of Int-C were indicated in this figure.*

We have attempted to improve our schematics, but it is quite difficult to make them realistic and informative at the same time. This is particularly problematic for the edge-on views. We have changed our IHF schematic to avoid confusion with which subunit is 'on top' (discussed more below). The IHF handedness in panel B was fixed and we have included an arrow to indicate the extra density that could be NTD-C. Note: Figure 5 is Figure 6 in the revised manuscript.

*Figure 6: It would help to add the axis of rotation relating the cartoons in A and B. It may not be obvious to everyone that the axis is perpendicular to the page, rather than parallel to it as it is the case for the structures. Also, the legend should specify that the structure in C corresponds to the cartoon viewed from the front.*

We have created new schematics for panels A and B, so the views of structures and cartoons are the same in each case. This was partly based on the suggestion above that we provide more realistic schematic models. Note: Figure 6 is Figure 7 in the revised manuscript.

*Figure 8: Every IHF in this figure has the wrong handedness (i.e. the wrong arm is in front of the DNA).*

Thank you for noticing this. We have also modified the schematics in A & B in a similar manner as for Figure 6 (new Figure 7), so they are more realistic. Note: Figure 8 is now Figure 10 in the revised manuscript.